# FuseCodec: Semantic-Contextual Fusion and Supervision for Neural Codecs

## Abstract

Speech tokenization enables discrete representation and facilitates speech language modeling. However, existing neural codecs capture low-level acoustic features, overlooking the semantic and contextual cues inherent to human speech. While recent efforts introduced semantic representations from self-supervised speech models or incorporated contextual representations from pre-trained language models, challenges remain in aligning and unifying the semantic and contextual representations. We introduce FuseCodec, which unifies acoustic, semantic, and contextual representations through strong cross-modal alignment and globally informed supervision. We propose three complementary techniques: (i) Latent Representation Fusion, integrating semantic and contextual features directly into the encoder latent space for robust and unified representation learning; (ii) Global Semantic-Contextual Supervision, supervising discrete tokens with globally pooled and broadcasted representations to enhance temporal consistency and cross-modal alignment; and (iii) Temporally Aligned Contextual Supervision, strengthening alignment by dynamically matching contextual and speech tokens within a local window for fine-grained token-level supervision. We further introduce FuseCodec-TTS, demonstrating our methodology's applicability to zero-shot speech synthesis. Empirically, FuseCodec achieves state-of-the-art performance in LibriSpeech, surpassing EnCodec, SpeechTokenizer, and DAC in transcription accuracy, perceptual quality, intelligibility, and speaker similarity. Results highlight the effectiveness of contextually and semantically guided tokenization for speech tokenization and downstream tasks.

## 1 Introduction

Tokenization is a cornerstone of natural language processing (NLP), enabling language models to represent text in discrete units for efficient autoregressive modeling and scalable downstream applications (Schmidt et al., 2024). Inspired by this paradigm, the speech domain has increasingly adopted neural codecs, popularized by Encodec (Défossez et al., 2022) and SoundStream (Zeghidour et al., 2022). Neural codecs tokenize speech using an encoder, residual vector quantizer, and decoder architecture, enabling modeling discrete representations suitable for modular extension to downstream tasks such as speech synthesis (Wang et al., 2023).

However, learning discrete speech representations is more challenging than text due to the continuous and multidimensional nature of speech (Ju et al., 2024). While neural codecs learn *acoustic representations* (waveform and low-level signal characteristics), they struggle to capture high-level semantics, requiring downstream models to adopt additional self-supervised masked language objectives to derive *semantic representations* (phonetic content and linguistic meaning) (Borsos et al., 2023). To bridge this gap, recent work incorporates semantic distillation from self-supervised speech models (Borsos et al., 2023; Zhang et al., 2024; Défossez et al., 2024), which improves both reconstruction quality and semantic awareness of learned tokens. Yet another fundamental aspect of human speech remains missing: speech is inherently grounded in context and surrounding cues (Brown et al., 2022). Discrete speech representations, lacking contextual grounding, fall short of capturing this essential attribute Hallap et al. (2023). While language models have demonstrated strong capabilities in modeling such contextual dependencies from text corpora (Devlin et al., 2019a; Peters et al., 2018), speech tokenizers have yet to fully leverage these capabilities. Although a recent neural codec (Ahasan et al., 2024) explored matching discrete speech representations with contextual

representations from a pre-trained language model, it falls short in effective cross-modal alignment, constraining the model's ability to fully unify semantic and contextual information.

Despite recent progress, three challenges remain. First, current approaches fail to jointly capture all three aspects of speech: acoustic (from neural codecs), semantic (from self-supervised speech models), and contextual (from language models). Prior work largely focuses on semantics, neglecting contextual grounding (Zhang et al., 2024; Défossez et al., 2024; Ye et al., 2024). Second, while a recent effort (Ahasan et al., 2024) attempts to integrate contextual representations, it lacks effective mechanisms for aligning text and speech modalities. Third, existing methods rely on similarity-based matching objectives, without directly integrating semantic and contextual information into the latent space, limiting coherence and downstream performance (Ji et al., 2025). Table 1 highlights these gaps, showing prior codecs are restricted to acoustic and partially semantic modeling, while our approach is the first to unify acoustic, semantic, and contextual aspects with direct integration and alignment.

| Model | A | S | C | Sim. | Direct. | Align. |
|---|---|---|---|---|---|---|
| Encodec | ✓ | ✗ | ✗ | ✗ | ✗ | ✗ |
| DAC | ✓ | ✗ | ✗ | ✗ | ✗ | ✗ |
| FACodec | ✓ | ✗ | ✗ | ✗ | ✗ | ✗ |
| BigCodec | ✓ | ✗ | ✗ | ✗ | ✗ | ✗ |
| StableCodec | ✓ | ✗ | ✗ | ✗ | ✗ | ✗ |
| WavTokenizer | ✓ | ✓ | ✗ | ✗ | ✗ | ✗ |
| SpeechTokenizer | ✓ | ✓ | ✗ | ✗ | ✗ | ✗ |
| Mimi | ✓ | ✓ | ✗ | ✗ | ✗ | ✗ |
| DM-Codec | ✓ | ✓ | ✓ | ✓ | ✗ | ✗ |
| **FuseCodec** | ✓ | ✓ | ✓ | ✓ | ✓ | ✓ |

Table 1: Codec comparison across key aspects. Most codecs capture only acoustic (A) and partially semantic (S) information with similarity-based supervision (Sim.), without contextual grounding (C), direct latent integration (Direct.), or modality alignment (Align.); our **FuseCodec** unifies all aspects.

To address these challenges, we propose three strategies that enrich discrete speech representations with unified semantic and contextual information: (i) **Latent Representation Fusion** (FuseCodec-Fusion) integrates semantic and contextual embeddings into the encoder's latent space through cross-modal attention and additive fusion, yielding more coherent representations. (ii) **Global Semantic-Contextual Supervision** (FuseCodec-Distill) uses globally pooled and broadcasted modality vectors to supervise each quantized token across time, ensuring temporally consistent and globally informed learning. (iii) **Temporally Aligned Contextual Supervision** (FuseCodec-ContextAlign) dynamically matches contextual and speech tokens prior to time step-level similarity supervision, enabling fine-grained cross-modal alignment and enhancing representation quality.

FuseCodec establishes state-of-the-art performance on LibriSpeech test set, outperforming En-Codec, SpeechTokenizer, and DM-Codec in both intelligibility and perceptual quality. On Codec-SUPERB, it delivers the best signal-level and strong downstream task performance, surpassing recent codecs such as DAC, BigCodec, and X-Codec2 while operating at only 4 kbps. Moreover, FuseCodec extends effectively to zero-shot speech synthesis, underscoring the value of unified semantic and contextual grounding in discrete speech tokenization.

Therefore, our key contributions are:

- We introduce a unified speech tokenization framework with three codec variants: FuseCodec-Fusion, FuseCodec-Distill, and FuseCodec-ContextAlign, integrating semantic and contextual information via latent fusion, global supervision, and temporal alignment.

- Our approach substantially improves speech reconstruction and representation quality, establishing new state-of-the-art results on LibriSpeech and outperforming prior codecs on the Codec-SUPERB benchmark.

- We validate the effectiveness of each component through extensive ablations and demonstrate practical utility in downstream text-to-speech generation.

## 2 FUSECODEC

As shown in Figure 1, we first introduce the speech discretization pipeline (§2.1) and describe the extraction of semantic and contextual representations from pre-trained models (§2.2). We then present three strategies for integrating multimodal guidance into speech tokenization: (i) Latent Representation Fusion (§2.3.1), (ii) Global Semantic-Contextual Supervision (§2.3.2), and (iii) Temporally Aligned Contextual Supervision (§2.3.3). Finally, we outline the training objective (§2.4) and the extension to a text-to-speech task (§2.5).

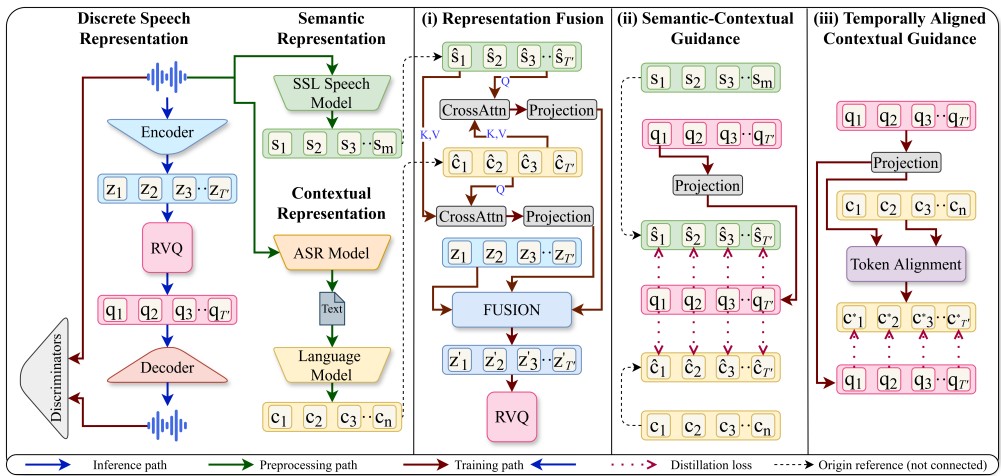

Figure 1: Overview of the FuseCodec speech tokenization framework. Input speech $\mathbf{x}$ is encoded into latent features $\mathbf{Z}$, then quantized into discrete tokens $\mathbf{Q}^{(1:K)}$ via residual vector quantization (RVQ). To enrich these tokens, we incorporate semantic ($\mathbf{S}_i, \hat{\mathbf{S}}$) and contextual ($\mathbf{C}_i, \hat{\mathbf{C}}, \mathbf{C}^*$) representations from frozen pre-trained models. Global vectors $\hat{\mathbf{S}}$ and $\hat{\mathbf{C}}$ are formed via mean pooling and [CLS] selection, respectively. We propose three strategies: (i) Latent Representation Fusion, injecting global vectors $\hat{\mathbf{S}}, \hat{\mathbf{C}}$ with $\mathbf{Z}$ to yield fused latent $\mathbf{Z}'$; (ii) Global Semantic-Contextual Supervision, supervising $\mathbf{Q}^{(1)}$ with global vectors; and (iii) Temporally Aligned Contextual Supervision, aligning full contextual embeddings $\{\mathbf{C}_i\}$ to RVQ outputs via a windowed matching algorithm to form $\mathbf{C}^*$.

## 2.1 DISCRETE SPEECH REPRESENTATION

Discrete tokens serve as the foundation of neural codec-based speech-language models. Following established approaches (Défossez et al., 2022; Zhang et al., 2024; Ahasan et al., 2024), we discretize audio using an encoder-quantizer setup.

Given an input speech waveform $\mathbf{x}$, an encoder $E$ compresses $\mathbf{x}$ into a sequence of latent representations $\mathbf{Z} = \{\mathbf{z}_i\}_{i=1}^{T'}$, where $T'$ is the number of encoded frames. The encoder output $\mathbf{Z}$ is then passed through a Residual Vector Quantization module (RVQ), consisting of $K$ quantization layers. For layer $k \in \{1, \ldots, K\}$, the RVQ produces a sequence of token indices $\{q_i^{(k)}\}_{i=1}^{T'}$. Each index $q_i^{(k)}$ is then mapped to its embedding in the $k$-th codebook, yielding the sequence of quantized vectors $\mathbf{Q}^{(k)} = \{\mathbf{q}_i^{(k)}\}_{i=1}^{T'}$, where $\mathbf{q}_i^{(k)} \in \mathbb{R}^D$ and $D$ denotes the embedding dimensionality.

## 2.2 MULTIMODAL REPRESENTATION EXTRACTION

Concurrently, we extract representations from pre-trained models. Specifically, we obtain contextual representations from a pre-trained language model, which are dynamic, token-level embeddings that adapt to surrounding text (Devlin et al., 2019b; Peters et al., 2018). In parallel, we derive semantic representations from a pre-trained self-supervised speech model, which capture the high-level structure and meaning (Borsos et al., 2023).

**Contextual Representation.** The input speech waveform $\mathbf{x}$ is transcribed into text $\mathbf{x}'$ using a pre-trained Automatic Speech Recognition (ASR) model $A$, such that $\mathbf{x}' = A(\mathbf{x})$. The ASR model functions purely as a speech-to-text converter and remains detached during training. The transcribed text $\mathbf{x}'$ is processed by a pre-trained language model $B$, which produces a token sequence $\{c_i\}_{i=1}^n$. For each token $c_i$, we extract hidden states from all $L$ layers, represented as $\{\mathbf{h}_i^{(l)}\}_{l=1}^L$. These are averaged to produce contextual embeddings: $\mathbf{C}_i = \frac{1}{L} \sum_{l=1}^L \mathbf{h}_i^{(l)}$, where $\mathbf{C}_i \in \mathbb{R}^{D'}$, and $D'$ denotes the hidden dimension of the language model.

**Semantic Representation.** The input speech waveform $\mathbf{x}$ is passed through a pre-trained self-supervised speech model $H$, which outputs a sequence of frame-level tokens $\{s_i\}_{i=1}^m$. For each frame $s_i$, we extract hidden states from all $L$ layers: $\{\mathbf{h}_i^{(l)}\}_{l=1}^L$. These are averaged to obtain semantic embeddings: $\mathbf{S}_i = \frac{1}{L} \sum_{l=1}^L \mathbf{h}_i^{(l)}$, where $\mathbf{S}_i \in \mathbb{R}^{D'}$, and $D'$ denotes the hidden dimension.

## 2.3 SEMANTIC-CONTEXTUAL GUIDANCE

Our goal is to enrich discrete speech representations by integrating contextual and semantic information, enabling tighter alignment between acoustic structure and linguistic meaning. Prior work has explored similar directions: Zhang et al. (2024); Défossez et al. (2024) aligned HuBERT-based semantic features with the first RVQ layer using cosine similarity, while Ahasan et al. (2024) matched BERT-based embeddings to RVQ outputs via padded sequences and similarity loss. However, these methods either rely on a single modality (semantic in Zhang et al. (2024); Défossez et al. (2024)) or lack robust cross-modal alignment (misaligned context in Ahasan et al. (2024)).

In contrast, we unify semantic and contextual representations while ensuring robust alignment. For this, we propose three strategies: (i) Latent Representation Fusion (§2.3.1), (ii) Global Semantic-Contextual Supervision (§2.3.2), and (iii) Temporally Aligned Contextual Supervision (§2.3.3)

### 2.3.1 LATENT REPRESENTATION FUSION

We first propose to fuse semantic and contextual representations with the encoder's latent representations. The enhanced latents are then passed to the residual vector quantization (RVQ) module, enabling the learning of discrete codes enriched with semantic and contextual information.

Specifically, we apply mean pooling over the semantic embeddings $\{\mathbf{S}_i\}_{i=1}^m$ to compute the global semantic vector $\hat{\mathbf{S}} = \frac{1}{m} \sum_{i=1}^m \mathbf{S}_i$. For the textual modality, we select the `[CLS]` token embedding from the contextual representations $\{\mathbf{C}_i\}_{i=1}^n$, yielding $\hat{\mathbf{C}} = \mathbf{C}_{[\text{CLS}]}$. We then broadcast each global vector across the discrete token sequence length $T'$, forming: $\tilde{\mathbf{S}} = \{\hat{\mathbf{S}}\}_{t=1}^{T'}$, and $\tilde{\mathbf{C}} = \{\hat{\mathbf{C}}\}_{t=1}^{T'}$. Broadcasting allows each token to inherit the full semantic or contextual knowledge of the sequence, ensuring every position is enriched with the most informative signal for cross-modal fusion or distillation. Next, we apply multi-head cross-attention to enable cross-modal interaction, followed by an MLP projection to match the encoder dimension $D$:

$$\mathbf{S}' = \text{CrossAttention}(\tilde{\mathbf{S}}, \tilde{\mathbf{C}}, \tilde{\mathbf{C}})\mathbf{W}_S, \quad \mathbf{C}' = \text{CrossAttention}(\tilde{\mathbf{C}}, \tilde{\mathbf{S}}, \tilde{\mathbf{S}})\mathbf{W}_C, \quad (1)$$

where $\mathbf{W}_S, \mathbf{W}_C \in \mathbb{R}^{D' \times D}$ are learned projection matrices and CrossAttention$(\cdot)$ denotes multi-head cross-attention. Finally, we fuse the modality signals with the latent representation $\mathbf{Z} \in \mathbb{R}^{T' \times D}$ via additive fusion and modality dropout:

$$\mathbf{Z}' = \mathbf{Z} + (\mathbf{S}' \odot \mathcal{D}_S) + (\mathbf{C}' \odot \mathcal{D}_C), \quad (2)$$

where $\mathcal{D}_S, \mathcal{D}_C \in \{0,1\}^{T' \times D}$ are stochastic dropout masks applied during training. Dropout promotes robustness by preventing the quantized representations from over-relying on the fused modalities (Hussen Abdelaziz et al., 2020), and allows inference using only the encoder signal. The resulting fused representation $\mathbf{Z}'$ is then passed to the RVQ module for discrete speech quantization.

### 2.3.2 GLOBAL SEMANTIC-CONTEXTUAL SUPERVISION

In addition to latent fusion, we introduce an alternative representation supervision strategy, motivated by its effectiveness of similarity matching in prior speech tokenization work (Zhang et al., 2024; Défossez et al., 2024; Ahasan et al., 2024). Existing methods typically constrain representations along feature dimensions or through local frame-level alignment, which limits temporal consistency. In contrast, we propose a global-to-local time-axis distillation scheme: global semantic ($\hat{\mathbf{S}}$) and contextual ($\hat{\mathbf{C}}$) vectors directly supervise the RVQ outputs across time, enforcing consistent temporal guidance and pushing the quantized space to capture modality-aware temporal dynamics.

Together with our global semantic–contextual supervision, we redefine the combined distillation loss of Ahasan et al. (2024) to operate along the temporal rather than the feature axis. By embedding global signals into every timestep, our approach achieves stronger cross-modal coherence, temporally robust discrete codes, and richer unification of semantic and contextual structure.

Given the broadcasted global signals (see 2.3.1) $\tilde{\mathbf{S}}, \tilde{\mathbf{C}} \in \mathbb{R}^{T' \times D'}$, we apply a linear projection to the first-layer RVQ output $\mathbf{Q}^{(1)} \in \mathbb{R}^{T' \times D}$ to align dimensionality: $\mathbf{Q}'^{(1)} = \mathbf{Q}^{(1)}\mathbf{W}$, where $\mathbf{W} \in \mathbb{R}^{D \times D'}$. We then apply the *semantic–contextual supervision loss*:

$$\mathcal{L}_{\text{distill}} = -\frac{1}{T'} \sum_{t=1}^{T'} \log \sigma \left( \frac{1}{2} [\cos(\mathbf{Q}'^{(1)}_t, \tilde{\mathbf{S}}_t) + \cos(\mathbf{Q}'^{(1)}_t, \tilde{\mathbf{C}}_t)] \right) \quad (3)$$

where $\sigma(\cdot)$ is the sigmoid function and $\cos(\cdot, \cdot)$ denotes cosine similarity. This formulation provides fine-grained temporal supervision using global modality signals, enhancing the representational quality of the learned discrete tokens.

### 2.3.3 TEMPORALLY ALIGNED CONTEXTUAL SUPERVISION

Building on our use of the global contextual vector $\hat{\mathbf{C}}$ for supervision, we propose a finer-grained approach that leverages the full sequence of contextual embeddings $\{\mathbf{C}_i\}_{i=1}^n$ to supervise the RVQ token sequence $\{\mathbf{q}_t^{(1)}\}_{t=1}^{T'}$, enabling richer, timestep-level guidance. A key challenge, however, is the mismatch in sequence lengths between the contextual embeddings ($n$) and the RVQ output ($T'$).

To address this, we introduce a *dynamic window-based alignment strategy* (Algorithm 1). For each contextual embedding $\mathbf{C}_i$, the method defines a localized search window of RVQ tokens: either evenly divided across the sequence or adaptively shifted based on the previous match. Within this window, we compute cosine similarities and assign $\mathbf{C}_i$ to the token(s) with maximum similarity. If multiple tokens achieve the maximum, the embedding is broadcast to all of them, capturing the frequent case where a single text token corresponds to multiple acoustic frame tokens. After each match, the search window shifts forward, ensuring coverage of the entire sequence without overlap or collapse. The resulting sequence $\mathbf{C}^* \in \mathbb{R}^{T' \times D'}$ serves as a temporally aligned supervision signal matched to RVQ tokens $\{\mathbf{Q}_t'^{(1)}\}_{t=1}^{T'}$ for the *aligned contextual supervision loss*, applied as:

$$\mathcal{L}_{\text{distill}} = -\frac{1}{T'} \sum_{t=1}^{T'} \log \sigma \left( \cos\left(\mathbf{Q}_t'^{(1)}, \mathbf{C}_t^*\right) \right) \quad (4)$$

where $\mathbf{Q}'^{(1)} = \mathbf{Q}^{(1)} \mathbf{W} \in \mathbb{R}^{T' \times D'}$ is the linearly projected RVQ output, and $\sigma(\cdot)$ denotes the sigmoid function. This loss enforces temporally precise alignment between RVQ tokens and their corresponding contextual representations.

---

**Algorithm 1: Window-Based Token Alignment**

---

**Require:** Contextual embeddings $\{\mathbf{C}_i\}_{i=1}^n$, RVQ tokens $\{\mathbf{Q}_t^{(1)}\}_{t=1}^{T'}$, optional window size $w$
1: **if** $w$ not provided **then**
2:    $w \leftarrow \lfloor T'/n \rfloor$
3: **end if**
4: Initialize aligned output $\mathbf{C}^* \in \mathbb{R}^{T' \times D'} \leftarrow 0$
5: Initialize $\ell \leftarrow 0$ {last matched index}
6: **for** $i = 1$ to $n$ **do**
7:    **if** dynamic window **then**
8:       $s \leftarrow \ell + 1$ if $i > 1$, else $0$ {start index}
9:       $e \leftarrow \min(s + w, T')$ {end index}
10:    **else**
11:       $s \leftarrow (i-1) \cdot w, e \leftarrow \min(s + w, T')$
12:    **end if**
13:    Compute cosine similarity
      $\alpha_t = \cos(\mathbf{C}_i, \mathbf{Q}_t^{(1)})$ for $t \in [s, e)$
14:    Let $\tau \leftarrow \max_t \alpha_t$ {maximum similarity}
15:    $\mathcal{T}_i \leftarrow \{t \mid \alpha_t \geq \tau\}$
16:    **for** each $t \in \mathcal{T}_i$ **do**
17:       $\mathbf{C}_t^* \leftarrow \mathbf{C}_i$
18:    **end for**
19:    $\ell \leftarrow \max(\mathcal{T}_i)$
20: **end for**
21: **return** $\mathbf{C}^*$

---

### 2.4 ARCHITECTURE AND TRAINING OBJECTIVE

We build on widely adopted neural codec architectures and training objectives, following (Défossez et al., 2022; Zhang et al., 2024; Ahasan et al., 2024), to establish a strong and reliable foundation. We contribute to enhancing the learned representations through semantic and contextual supervision and fusion without altering the model architecture.

**Architecture.** We use wav2vec 2.0 (base-960h) as the ASR model $A$ (Baevski et al., 2020), BERT (bert-base-uncased) as the language model $B$ (Devlin et al., 2019a), and HuBERT (base-ls960) as the self-supervised speech model $H$ (Hsu et al., 2021). All pre-trained models are frozen during training. The speech tokenizer consists of an encoder $E$, an RVQ module with 8 quantization layers (codebooks) of size 1024, a decoder $D$, and three discriminators (multi-period, multi-scale, and multi-scale STFT). Architectural details are provided in Sec. E.1. Quantization operates on 50 Hz frame rates. The encoder and RVQ use an embedding dimension of $D = 1024$, while the pre-trained langauge and speech model have $D' = 768$. Cross-Attentions are implemented using 8-heads. The dropout masks $\mathcal{D}_S$ and $\mathcal{D}_C$ are applied at a rate of 10%.

**Training Objective.** We also adopt a multi-objective training setup grounded in established neural codec practices. This includes time-domain reconstruction loss $\mathcal{L}_{\text{time}}$, frequency-domain reconstruction loss $\mathcal{L}_{\text{freq}}$, adversarial loss $\mathcal{L}_{\text{gen}}$, feature matching loss $\mathcal{L}_{\text{feat}}$, and RVQ commitment loss $\mathcal{L}_{\text{commit}}$

(see Sec. E.2 for details). For our proposed semantic-contextual fusion and supervision, the applied loss depends on the model variant: when training FuseCodec-Distill we use the semantic–contextual supervision loss as $\mathcal{L}_{\text{distill}}$ (Sec. 2.3.2); when training FuseCodec-ContextAlign we use the aligned contextual supervision loss as $\mathcal{L}_{\text{distill}}$ (Sec. 2.3.3); and when training FuseCodec-Fusion (Sec. 2.3.1) both are disabled, with $\mathcal{L}_{\text{distill}} = 0$. The final training objective is a weighted sum:

$$\mathcal{L}_{\text{total}} = \lambda_{\text{time}}\mathcal{L}_{\text{time}} + \lambda_{\text{freq}}\mathcal{L}_{\text{freq}} + \lambda_{\text{gen}}\mathcal{L}_{\text{gen}} + \lambda_{\text{feat}}\mathcal{L}_{\text{feat}} + \lambda_{\text{commit}}\mathcal{L}_{\text{commit}} + \big(\lambda_{\text{distill}}\mathcal{L}_{\text{distill}} \text{ or } 0\big) \quad (5)$$

## 2.5 DOWNSTREAM EXTENSION TO TTS MODEL

We extend the learned discrete token representations to a downstream text-to-speech (TTS) task, following the neural codec language modeling framework and objective used in prior work (Wang et al., 2023; Zhang et al., 2024; Ahasan et al., 2024). In this paradigm, speech synthesis is performed by predicting quantized acoustic tokens produced by the RVQ and decoded by a neural codec. We extend the learned discrete tokens to TTS, with variants inheriting each fusion or supervision strategy, enabling synthesis from tokens that capture acoustic, semantic, and contextual information.

Given a phoneme sequence $\mathbf{p}$ and an acoustic prompt $\mathbf{A} \in \mathbb{R}^{\tau \times K}$ extracted from a reference utterance using FuseCodec, we predict discrete token indices $q^{(1)}, \ldots, q^{(K)}$ for the $K$ RVQ layers.

To model coarse content and prosody, the first-layer tokens $q^{(1)}$ are predicted autoregressively with a decoder-only Transformer conditioned on $\mathbf{p}$, using the objective:

$$\mathcal{L}_{\text{AR}} = -\log \prod_{i=1}^{T'} p\big(q_i^{(1)} \mid q_{<i}^{(1)}, \mathbf{p}; \theta_{\text{AR}}\big) \quad (6)$$

For fine-grained acoustic details, higher-layer tokens $q^{(k)}$ ($k = 2, \ldots, K$) are predicted non-autoregressively conditioned on $q^{(<k)}$, $\mathbf{p}$, and $\mathbf{A}$:

$$\mathcal{L}_{\text{NAR}} = -\log \prod_{k=2}^{K} p\big(q^{(k)} \mid q^{(<k)}, \mathbf{p}, \mathbf{A}; \theta_{\text{NAR}}\big) \quad (7)$$

Both AR and NAR models use 12-layer Transformers with 16 attention heads, 1024-dim embeddings, 4096-dim feed-forward layers, and 0.1 dropout. Predicted tokens are mapped to embeddings $\mathbf{Q}^{(k)}$ and decoded by FuseCodec to synthesize speech.

## 3 EXPERIMENTS

We describe our experimental setup (§3.1) and present main results and ablation studies (§3.2–§3.3).

## 3.1 EXPERIMENTAL SETUP

**Training.** Following prior work in speech tokenization (Zhang et al., 2024; Ahasan et al., 2024), we train FuseCodec on the LibriSpeech (Panayotov et al., 2015) train-clean-100 subset, which contains 100 hours of English speech from 251 speakers, sampled at 16 kHz. During training, we randomly crop 3-second audio segments and reserve 100 samples for validation. For FuseCodec-TTS, we combine the train and dev subsets of LibriTTS (Zen et al., 2019), comprising 570 hours of speech. FuseCodec is trained for 100 epochs on two A40 GPUs with a batch size of 6, using the Adam optimizer with a learning rate of $1 \times 10^{-4}$ and exponential decay factor 0.98. FuseCodec-TTS is trained on A100 and L40S GPUs. The AR model is trained for 200 epochs, and the NAR model for 150 epochs. Training employs dynamic batching, with each batch containing up to 550 seconds of audio for AR and 100–200 seconds for NAR. We use the ScaledAdam optimizer with a learning rate of $5 \times 10^{-2}$ and 200 warm-up steps.

**Baselines.** We compare FuseCodec against both established and recent strong baseline speech tokenizers, including EnCodec (Défossez et al., 2022) and SpeechTokenizer (Zhang et al., 2024), BigCodec (Xin et al., 2024), DAC (Kumar et al., 2023), DM-Codec (LM+SM) (Ahasan et al., 2024) FACodec (NaturalSpeech 3) (Ju et al., 2024), Moshi (Défossez et al., 2024), StableCodec (Parker et al., 2025), WavTokenize (Ji et al., 2025), and X-codec2 (Ye et al., 2025). All baseline results are obtained using official released checkpoints. For FuseCodec-TTS, we compare with neural codec language models that incorporate external representation guidance. Specifically, we compare against USLM (from SpeechTokenizer) (Zhang et al., 2024) and DM-Codec-TTS (Ahasan et al., 2024), using their official released LibriTTS trained checkpoints.

**Metrics.** We evaluate FuseCodec on: *Content Preservation* and *Speech Naturalness*. For *Content Preservation*, generated speech is transcribed with Whisper (medium) (Radford et al., 2023) and compared to the reference. We report *Word Error Rate (WER)*: WER $= \frac{S+D+I}{N}$, with $S$, $D$, $I$ as substitutions, deletions, insertions, and $N$ the reference word count. *Word Information Lost (WIL)* is WIL $= 1 - \frac{C}{N} + \frac{C}{P}$, where $C$ is correct words and $P$ predicted words. *Short-Time Objective Intelligibility (STOI)* estimates intelligibility via short-time spectral similarity. For *Speech Naturalness*, we assess perceptual and acoustic fidelity using reference-based and learned metrics. *ViSQOL* and *PESQ* model auditory similarity and signal distortion, respectively. *UTMOS* predicts human-judged naturalness, and *Similarity* computes cosine similarity between L2-normalized WavLM-TDNN embeddings (Chen et al., 2022) to measure speaker or content consistency. For FuseCodec-TTS, reference-based metrics (STOI, ViSQOL, PESQ) are omitted since references are unavailable.

## 3.2 MAIN RESULTS

We evaluate FuseCodec variants on speech reconstruction (§3.2.1), representation quality (§3.2.2), and downstream speech generation (§3.2.3).

### 3.2.1 SPEECH RECONSTRUCTION EVALUATION

Table 2: **Speech reconstruction results** on content preservation and naturalness metrics using various codecs. **Bold** highlights best scores, and underline indicates our second-best scores. Bw = bandwidth in kbps, Nq = number of quantizers, FR = frame rate in Hz. Overall, **FuseCodec variants consistently achieve strong reconstruction performance by unifying semantic and contextual information in discrete representations.**

| Model | Config (Bw/Nq/FR) | Content Preservation | | | Speech Naturalness | | | |
|---|---|---|---|---|---|---|---|---|
| | | WER↓ | WIL↓ | STOI↑ | ViSQOL↑ | PESQ↑ | UTMOS↑ | Similarity↑ |
| BigCodec | 1.04 / 8 / 50 | 4.58 | 7.45 | 0.93 | 3.02 | 2.68 | 3.44 | 0.996 |
| DAC | 6 / 12 / 50 | 4.09 | 6.54 | 0.94 | 3.36 | 2.72 | 3.33 | 0.996 |
| DM-Codec | 4 / 8 / 50 | 4.09 | 6.75 | 0.93 | 3.20 | 2.77 | 3.45 | 0.994 |
| EnCodec | 6 / 8 / 75 | 4.04 | 6.58 | 0.92 | 3.06 | 2.31 | 2.41 | 0.980 |
| FACodec | 4.8 / 6 / 80 | 4.11 | 6.58 | **0.95** | 3.11 | 2.89 | 3.45 | 0.996 |
| Mimi | 1.1 / 8 / 12.5 | 11.61 | 18.05 | 0.85 | 2.49 | 1.69 | 2.28 | 0.934 |
| SpeechTokenizer | 4 / 8 / 50 | 4.16 | 6.71 | 0.92 | 3.08 | 2.60 | 3.41 | 0.996 |
| StableCodec | 0.625 / 6 / 25 | 10.32 | 15.87 | 0.88 | 2.51 | 1.95 | 3.58 | 0.984 |
| WavTokenizer | 0.9 / 1 / 75 | 6.28 | 10.11 | 0.89 | 2.59 | 2.13 | 3.36 | 0.993 |
| X-codec2 | 0.8 / 1 / 50 | 4.46 | 7.20 | 0.92 | 2.87 | 2.43 | 3.55 | **0.997** |
| FuseCodec (Baseline) | 6 / 8 / 50 | 4.62 | 7.44 | 0.93 | 2.95 | 2.54 | 3.18 | 0.990 |
| FuseCodec-ContextAlign | 4 / 8 / 50 | 4.15 | 6.70 | 0.93 | 3.18 | 2.85 | **3.65** | 0.995 |
| FuseCodec-Distill | 4 / 8 / 50 | 4.09 | 6.60 | 0.94 | 3.43 | 3.06 | **3.65** | 0.996 |
| FuseCodec-Fusion | 4 / 8 / 50 | **3.99** | **6.45** | **0.95** | **3.47** | **3.13** | 3.63 | 0.995 |

This evaluation measures how well FuseCodec preserves both linguistic content and perceptual quality in speech reconstruction. We compare against widely used and trending codecs, selecting model configurations that closely match ours for fairness. Consistent with established practice, we evaluate on the LibriSpeech test-clean subset (2620 utterances), which has been the standard and exclusive benchmark in prior neural codec studies (Zhang et al., 2024; Ahasan et al., 2024; Xin et al., 2024; Défossez et al., 2024; Parker et al., 2025; Ye et al., 2024; 2025). Table 2 reports the results, revealing:

*(i) Best overall.* **FuseCodec-Fusion consistently achieves the strongest reconstruction performance.** It records the lowest WER (3.99) and WIL (6.45), along with the highest STOI (0.95), ViSQOL (3.47), and PESQ (3.13). Compared to EnCodec, which models only acoustics, and FA-Codec, which separates attributes without unifying them, FuseCodec-Fusion integrates both semantic and contextual signals directly into the encoder's latent space. This unified representation improves intelligibility and perceptual quality, also outperforming compression-focused models such as DAC, BigCodec, StableCodec, WavTokenizer, and X-Codec2.

*(ii) Naturalness and speaker consistency.* **FuseCodec-Distill excels in perceptual quality and speaker similarity**, achieving the top UTMOS (3.65) and Similarity (0.996), while ranking second in STOI (0.94), ViSQOL (3.43), and PESQ (3.06). It surpasses models such as SpeechTokenizer, X-Codec2, and Mimi, which capture only semantic signals, as well as codecs lacking supervision: EnCodec, DAC, StableCodec, and WavTokenizer. By supervising the quantized space with global semantic and contextual signals, FuseCodec-Distill aligns discrete tokens with both linguistic and acoustic content, yielding natural and consistent speech.

*(iii) Interpretable local alignment.* **FuseCodec-ContextAlign delivers competitive performance with aligned token-level supervision.** It matches the top UTMOS (3.65) and improves over the

baseline FuseCodec (Baseline) across all metrics, outperforming BigCodec, Mimi, SpeechTokenizer, StableCodec, WavTokenizer, and X-Codec2 with lower WER (6.70), WIL (4.15), and higher STOI (0.94), ViSQOL (3.18), and PESQ (2.85). These gains show that aligning discrete tokens with contextual information strengthens local content preservation and enhances intelligibility. Although its constrained alignment limits global contextual guidance, yielding slightly lower performance than FuseCodec-Fusion and FuseCodec-Distill. Taken together, these results show that integrating semantic and contextual signals in the latent space substantially improves speech reconstruction.

### 3.2.2 REPRESENTATION QUALITY EVALUATION

Table 3: **Representation quality results** on the `Codec-SUPERB` benchmark. Signal-level evaluation: **Audio** (Mel, STFT) and **Speech** (PESQ, STOI, F0CORR) metrics. Application-level evaluation: **ASR** = automatic speech recognition, **ASV** = speaker verification, **ER** = emotion recognition, **AEC** = audio event classification. **Bold** highlights the best scores, and underline indicates our second-best scores. Overall, **FuseCodec variants achieve top performance across both signal-level and downstream tasks, demonstrating effective latent representations at low bitrates.**

| (a) Codec Information | | | (b) Signal-level | | (c) Application-level | | | |
|---|---|---|---|---|---|---|---|---|
| **Model** | **kbps** | **Other Configuration** | **Speech↑** | **Audio↑** | **ASR↓** | **ASV↓** | **ER↑** | **AEC↑** |
| None | - | - | - | - | 2.96 | 0.86 | 69.84 | 45.68 |
| SpeechTokenizer | 4 | 16k | 0.644 | 0.581 | 4.02 | 3.31 | 65.49 | 15.11 |
| AcademiCodec | 2 | 16k_320d | 0.610 | 0.574 | 4.94 | 4.43 | 65.96 | 16.19 |
| AcademiCodec | 2 | 16k_320d_large_uni | 0.617 | 0.574 | 6.26 | 5.22 | 64.63 | 28.65 |
| AcademiCodec | 3 | 24k_320d | 0.611 | 0.592 | 4.49 | 6.16 | 65.95 | 14.01 |
| AudioDec | 6.4 | 24k_320d | 0.596 | 0.602 | 3.94 | 5.22 | 65.70 | 17.41 |
| DAC | 6 | 16k | **0.798** | 0.591 | **3.26** | **1.59** | 68.81 | 41.08 |
| EnCodec | 1.5 | 24k | 0.579 | 0.594 | 9.21 | 13.88 | 58.84 | 18.84 |
| EnCodec | 3 | 24k | 0.636 | 0.599 | 4.34 | 6.85 | 63.54 | 26.63 |
| EnCodec | 6 | 24k | 0.697 | 0.602 | 3.49 | 4.28 | 66.18 | 32.43 |
| FunCodec | 8 | en_libritts_16k_nq32ds640 | 0.678 | 0.578 | 3.43 | 2.04 | 68.26 | 21.43 |
| FunCodec | 8 | zh_en_16k_nq32ds640 | 0.718 | 0.583 | 3.27 | 1.60 | 69.55 | 33.59 |
| FuseCodec-ContextAlign | 4 | 16k | 0.698 | 0.771 | 4.24 | 3.40 | 73.19 | 57.20 |
| FuseCodec-Distill | 4 | 16k | 0.731 | 0.784 | 3.38 | 3.12 | 73.82 | **57.25** |
| FuseCodec-Fusion | 4 | 16k | 0.744 | **0.785** | 3.44 | 3.85 | **73.96** | 55.35 |

To assess the representational quality of FuseCodec beyond reconstruction, we conduct experiments on the `Codec-SUPERB` benchmark (Wu et al., 2024). The benchmark comprises application-level tasks: *automatic speech recognition (ASR)*; *automatic speaker verification (ASV)*; *emotion recognition (ER)*; and *audio event classification (AEC)*. Signal-level evaluation is reported separately for *audio* (Mel, STFT) and *speech* (PESQ, STOI, F0CORR). For fair comparison, we report results from (Wu et al., 2024), selecting models with configurations aligned to ours (4 kbps, 16 kHz). Baselines with higher bandwidths (≥8 kbps) or sampling rates (>24 kHz) are excluded, as their advantage comes from greater information capacity rather than method design. The music metric is omitted, as it lies outside our scope. Table 3 presents the evaluation results, highlighting:

*(i) High-quality speech and audio signals.* **FuseCodec-Fusion achieves the highest signal-level performance,** with the top Audio score (0.785) and second-highest Speech score (0.744), outperforming SpeechTokenizer, AudioDec, FunCodec, EnCodec, and AcademiCodec. Additionally, **FuseCodec-Distill and FuseCodec-ContextAlign maintain strong signal-level quality**, with Distill at 0.784 Audio and 0.731 Speech, and ContextAlign at 0.771 Audio and 0.698 Speech, showing that FuseCodec improves signal quality through semantic and contextual information retention.

*(ii) Downstream application generalization.* **FuseCodec variants excel on multiple downstream tasks,** showing strong generalization beyond speech reconstruction. Specifically, FuseCodec-Distill attains the best Audio Event Classification performance (57.25), while FuseCodec-Fusion achieves the highest Emotion Recognition accuracy (73.96). These results indicate that the representations learned by FuseCodec effectively capture task-relevant information, enabling superior performance on ER and AEC, despite FuseCodec being trained primarily for reconstruction.

*(iii) Balanced performance at lower bitrate.* While DAC achieves a slightly lower ASR error (3.26) and FunCodec reaches the lowest ASV error (1.60), **FuseCodec variants provide consistently strong performance across all metrics at only 4 kbps**, substantially lower than DAC (6 kbps) and FunCodec (8 kbps). This efficiency makes FuseCodec particularly well-suited for real-world speech applications, where reduced bitrates allow faster streaming, lower latency, and high-quality audio.

Table 4: **Speech generation results** on LibriSpeech and VCTK using zero-shot TTS. FuseCodec-TTS variants are compared with official neural codec-based TTS checkpoints trained on LibriTTS. **Bold** highlights best scores, and underline indicates second-best scores. Overall, **FuseCodec-Distill-TTS achieves the strongest intelligibility, FuseCodec-ContextAlign-TTS leads in naturalness, and FuseCodec-Fusion-TTS provides a well-rounded trade-off.**

| Model | WER ↓ | | WIL ↓ | | Similarity ↑ | | UTMOS ↑ | |
|---|---|---|---|---|---|---|---|---|
| | LibriSpeech | VCTK | LibriSpeech | VCTK | LibriSpeech | VCTK | LibriSpeech | VCTK |
| USLM | 16.72 | 14.79 | 25.65 | 23.24 | 0.80 | 0.78 | 2.93 | 3.01 |
| DM-Codec-TTS | 10.26 | 5.02 | 13.79 | 8.21 | 0.82 | **0.79** | 3.70 | 3.86 |
| FuseCodec-ContextAlign-TTS | 12.43 | 4.27 | 16.92 | 6.89 | **0.83** | 0.79 | **3.86** | **3.96** |
| FuseCodec-Distill-TTS | **8.55** | **3.66** | **12.07** | **6.02** | 0.82 | 0.78 | 3.55 | 3.75 |
| FuseCodec-Fusion-TTS | 9.67 | 4.07 | 13.23 | 7.18 | **0.83** | **0.79** | 3.63 | 3.82 |

### 3.2.3 DOWNSTREAM SPEECH GENERATION EVALUATION

We further evaluate the downstream extensibility of all FuseCodec variants on zero-shot TTS. Our goal is not to build the strongest TTS model, which is beyond our scope and resources, but to train on the smaller LibriTTS dataset and compare fairly with open-source codec models (e.g., SpeechTokenizer, DM-Codec) distilling representation. For evaluation, we adopt two established benchmarks. On LibriSpeech, following Wang et al. (2023), we select utterances 4–10 seconds long from test set, using 3-second enrollment segment from a different utterance of the same speaker. On VCTK, following Zhang et al. (2024), we use 3-second prompts from one utterance with transcript of another utterance by the same speaker as target. Table 4 presents the results, demonstrating:

*(i) Linguistic precision.* **FuseCodec-Distill-TTS leads in content preservation and intelligibility**, achieving the lowest WER (8.55 / 3.66) and WIL (12.07 / 6.02) on LibriSpeech and VCTK, and second-best similarity (0.82 / 0.78). Unlike USLM, which lacks contextual grounding, and DM-Codec-TTS, with limited context alignment, it distills global semantic-contextual representations into quantized tokens, enhancing both semantic and acoustic information.

*(ii) Perceptual quality.* **FuseCodec-ContextAlign-TTS delivers the highest perceptual naturalness**, achieving the best UTMOS scores (3.86 / 3.96) while also tying for top speaker similarity (0.83 / 0.79). Its temporally aligned contextual supervision enhances prosody modeling and speaker identity retention, clearly outperforming DM-Codec-TTS and USLM.

*(iii) Balanced performance.* **FuseCodec-Fusion-TTS offers the most balanced trade-off**, attaining joint-best similarity (0.83 / 0.79), competitive UTMOS (3.63 / 3.82), and solid intelligibility with second-best WER/WIL. Unlike DM-Codec-TTS, which lacks alignment, and USLM, which relies only on semantic features, FuseCodec-Fusion jointly integrates both semantic and contextual signals directly in the latent space, enabling synthesis that is both accurate and natural.

### 3.3 ADDITIONAL AND ABLATION STUDY RESULTS

**Unseen Multilingual Speech Reconstruction Evaluation.** We test FuseCodec on speech reconstruction across seven unseen languages (Appendix C.1). **FuseCodec-Fusion achieves the strongest content and perceptual scores**, with Distill maintaining second-best performance. Results show that FuseCodec generalizes robustly through unified semantic and contextual signals.

**Ablation Study.** We validate the design of FuseCodec through ablations (Appendix D). Key insights: *(i) Attention-projection:* cross-before yields best intelligibility and perceptual quality (See D.1); *(ii) Semantic-contextual guidance:* distilling both signals stabilizes tokens (See D.2); *(iii) Temporal alignment:* dynamic alignment improves clarity and content (See D.3); *(iv) Dropout:* 10% balances robustness and informativeness (See D.4); *(v) Quantizer supervision:* first-layer supervision strengthens semantic-contextual grounding (See D.5).

## 4 CONCLUSION

We introduced FuseCodec, a unified speech tokenization framework that integrates acoustic, semantic, and contextual signals via multimodal representation fusion and supervision. Our methods enable fine-grained alignment and achieve state-of-the-art results on speech reconstruction, improving intelligibility, quality, and speaker similarity. These findings highlight the value of semantic and contextual grounding in discrete speech modeling.

## 5 REPRODUCIBILITY STATEMENT

We ensure the reproducibility of our proposed FuseCodec and experimental results. The experimental setup, including datasets, training configurations, and hyperparameters, is described in Section 3.1. To facilitate replication, we provide links to anonymized resources in Appendix A, including a Docker environment, the full codebase, and trained model checkpoints, and include Python scripts for preprocessing and training, along with all necessary dependencies.

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

# Technical Appendix

## A    RESOURCES

We provide all necessary resources to ensure full reproducibility of our models and results. All links are anonymized for double-blind peer review.

- **Docker:** A containerized environment with all required Python packages for training. LINK
- **Code and Configuration:** Full codebase for preprocessing, training, and inference. LINK
- **Model Checkpoints:** Trained model weights. LINK

## B    RELATED WORK

Recent progress in speech and audio generation has been largely driven by advances in discrete representation learning, neural audio codecs, and language model-based synthesis. VQ-VAE (van den Oord et al., 2018) introduced vector quantization in latent spaces to support symbolic modeling of audio, while HuBERT (Hsu et al., 2021) applied masked prediction over cluster-derived labels to learn speech features in a self-supervised manner. SoundStream (Zeghidour et al., 2022) proposed a causal adversarially trained codec with residual vector quantization (RVQ) and demonstrated scalable compression at low bitrates. HiFi-Codec (Yang et al., 2023) further improved efficiency by introducing group residual quantization, reducing the number of required codebooks while preserving audio fidelity. On the generative side, AudioLM (Borsos et al., 2023) modeled long-range dependencies in semantic and acoustic tokens using transformer-based language modeling. This approach was extended by VALL-E (Wang et al., 2023), which enabled zero-shot text-to-speech synthesis by conditioning on short acoustic prompts and leveraging codec token generation. To improve the suitability of tokenization for language modeling tasks, X-Codec (Ye et al., 2024) integrated speech embeddings from pretrained models into the quantization pipeline, while LAST (Turetzky & Adi, 2024) learned a tokenizer supervised by a language model to improve downstream ASR and speech generation performance. HiFi-GAN (Kong et al., 2020) introduced multi-period and multi-scale discriminators, enabling high-fidelity waveform synthesis with real-time efficiency.

In parallel, codec designs have evolved to improve training stability and perceptual quality. En-Codec (Défossez et al., 2022) introduced a GAN-based codec architecture with multi-loss balancing and spectrogram-based discrimination, setting a new benchmark for real-time low-bitrate synthesis. BigCodec (Xin et al., 2024) scaled the VQ-VAE framework and showed that a single large codebook could achieve near-human perceptual quality at 1 kbps. DAC (Kumar et al., 2023) proposed refinements to residual quantization, such as factorized and normalized codebooks, and introduced advanced discriminators to improve quality under bitrate constraints. More recent work has focused on improving token expressiveness for downstream tasks. SpeechTokenizer (Zhang et al., 2024) demonstrated that hierarchical quantization improves reconstruction and zero-shot TTS, while DM-Codec (Ahasan et al., 2024) matched quantization layer representations with pre-trained speech and text models to reduce WER and enhance contextual fidelity. Finally, NaturalSpeech 3 (Ju et al., 2024) introduced a factorized codec to disentangle prosodic and acoustic attributes in speech, and Moshi (Défossez et al., 2024) unified ASR and TTS in a streaming, full-duplex transformer model operating on jointly learned speech tokens.

## C    ADDITIONAL RESULTS

We provide additional results on FuseCodec variants for multilingual speech reconstruction (§C.1).

### C.1    EXTENSION TO UNSEEN MULTILINGUAL SPEECH RECONSTRUCTION

This evaluation examines how well FuseCodec generalizes to unseen languages, testing whether integrating semantic and contextual signals in the latent space enables the codec to capture language-agnostic paralinguistic information. We use the Multilingual LibriSpeech test set (Pratap et al., 2020), covering German, Dutch, Spanish, French, Italian, Portuguese, and Polish. For fair comparison, we select baselines with multi-quantizer architectures, 16–24 kHz sampling, and 4–6 bit

Table 5: **Multilingual speech reconstruction results** across content preservation and perceptual metrics for unseen languages. **Bold** highlights the best score per language/metric, and underline indicates our second-best. Abbreviations: **nl** = Dutch, **fr** = French, **de** = German, **it** = Italian, **pl** = Polish, **pt** = Portuguese, **es** = Spanish. Overall, **FuseCodec variants maintain high content fidelity and perceptual quality across diverse languages by integrating semantic and contextual signals in the latent space.**

| Model | WER↓ | | | | | | | WIL↓ | | | | | | | PESQ↑ | | | | | | | VISQOL↑ | | | | | | |
|---|---|---|---|---|---|---|---|---|---|---|---|---|---|---|---|---|---|---|---|---|---|---|---|---|---|---|---|---|
| | nl | fr | de | it | pl | pt | es | nl | fr | de | it | pl | pt | es | nl | fr | de | it | pl | pt | es | nl | fr | de | it | pl | pt | es |
| SpeechTokenizer | 7.89 | 7.96 | 7.19 | 12.24 | 9.09 | 13.29 | 5.47 | 13.47 | 12.95 | 11.65 | 19.35 | 15.03 | 19.29 | 8.56 | 2.53 | 2.42 | 2.37 | 2.36 | 2.36 | 2.18 | 2.36 | 3.03 | 2.96 | 2.96 | 3.01 | 2.92 | 2.88 | 2.98 |
| EnCodec | 6.22 | 5.34 | 7.41 | 8.76 | 6.06 | 9.9 | **2.82** | 10.73 | 8.61 | 10.76 | 14.02 | 9.65 | 14.5 | **4.66** | 2.26 | 2.35 | 2.31 | 2.37 | 2.42 | 2.27 | 2.27 | 3.02 | 3.12 | 3.06 | 3.16 | 3.1 | 3.05 | 3.08 |
| DM-Codec | 6.94 | 6.36 | 5.93 | 10.34 | 6.7 | 11.69 | 4.69 | 11.85 | 10.53 | 9.76 | 16.56 | 11.45 | 16.45 | 7.12 | 2.83 | 2.65 | 2.57 | 2.66 | 2.65 | 2.36 | 2.61 | 3.19 | 3.15 | 3.15 | 3.22 | 3.15 | 3.07 | 3.18 |
| FaCodec | **5.34** | 5.98 | **4.82** | 8.89 | 5.22 | 9.84 | 3.26 | **9.23** | 9.87 | 8.16 | 14.09 | 8.9 | 14.57 | 5.5 | 2.80 | 2.68 | 2.63 | 2.63 | 2.76 | 2.46 | 2.62 | 3.13 | 2.99 | 3.01 | 3.02 | 3.05 | 2.94 | 3.03 |
| FuseCodec-Fusion | 5.80 | 6.92 | **4.82** | **7.97** | **5.07** | 8.75 | 3.53 | 9.77 | 8.40 | **7.84** | **12.95** | **8.63** | 13.09 | 5.20 | **3.15** | **3.04** | **2.92** | **3.03** | **3.11** | **2.82** | **2.94** | **3.46** | **3.43** | **3.42** | **3.48** | **3.42** | **3.38** | **3.43** |
| FuseCodec-Distill | 5.50 | **4.22** | 6.65 | 9.21 | 5.23 | **8.53** | 3.87 | 9.35 | **7.25** | 9.58 | 14.67 | 8.72 | **12.22** | 5.71 | 3.08 | 2.95 | 2.86 | 2.97 | 3.03 | 2.76 | 2.88 | 3.41 | 3.39 | 3.38 | 3.45 | 3.41 | 3.34 | 3.40 |
| FuseCodec-ContextAlign | 6.37 | 5.46 | 8.31 | 10.15 | 6.43 | 11.18 | 3.70 | 10.90 | 9.18 | 12.24 | 16.24 | 10.87 | 16.06 | 6.09 | 2.89 | 2.75 | 2.66 | 2.76 | 2.77 | 2.52 | 2.68 | 3.19 | 3.16 | 3.16 | 3.24 | 3.17 | 3.13 | 3.18 |

configurations, including SpeechTokenizer, EnCodec, DM-Codec, and FaCodec. Table 5 presents the results, revealing:

*(i) Content preservation.* **FuseCodec-Fusion achieves the lowest WER and WIL in three languages and ties for best WER and WIL in Portuguese,** while FuseCodec-Distill attains the best WER in French and Portuguese and second-best in Dutch. Across all seven languages, FuseCodec variants consistently rank first or second, whereas FaCodec and EnCodec win only in isolated cases. These results indicate that FuseCodec effectively retains core linguistic content and generalizes across diverse languages by unifying semantic and contextual signals.

*(ii) Perceptual quality.* **FuseCodec-Fusion delivers the highest PESQ and ViSQOL scores across all languages,** with Distill consistently second-best. Baselines trail by a substantial margin (Fusion improves PESQ by 0.3 or more over the next best model). This demonstrates that integrating semantic-contextual signals enhances perceptual naturalness and speech intelligibility, even in languages unseen during training.

*(iii) Cross-lingual robustness.* **FuseCodec-ContextAlign remains competitive, outperforming several baselines,** despite slightly lower performance than Fusion and Distill. It shows particular strengths on perceptual metrics (PESQ and ViSQOL) in Dutch, French, and Polish languages, often surpassing DM-Codec and SpeechTokenizer, which lack temporally aligned contextual supervision. Taken together, these results demonstrate that FuseCodec maintains high content accuracy and perceptual quality across unseen languages by unifying semantic and contextual representations.

# D   ABLATION STUDIES

We ablate and investigate each design choice and the necessity of components in our proposed methodology for FuseCodec. All model hyperparameters, training procedures, and configurations are kept fixed, except for the specific changes introduced in each ablation setup.

## D.1   ABLATION: ATTENTION-PROJECTION CONFIGURATION IN REPRESENTATION FUSION

Table 6: Ablation of attention-projection configurations in multimodal latent fusion. **Cross** variants incorporate cross-modal attention between semantic and contextual signals, while **Self** variants apply self-attention. **Before** applies attention prior to projection into the encoder's latent space, whereas **After** applies attention post-projection. **None** uses direct projection without attention. *Applying cross-modal attention before projection consistently improves content preservation and speech naturalness by enabling richer multimodal interactions in the original dimension.*

| Model Variant | Attn-Proj Type | Content Preservation | | | Speech Naturalness | | | |
|---|---|---|---|---|---|---|---|---|
| | | WER↓ | WIL↓ | STOI↑ | ViSQOL↑ | PESQ↑ | UTMOS↑ | Similarity↑ |
| FuseCodec-Fusion | None | 4.10 | 6.60 | 0.93 | 3.26 | 2.92 | **3.65** | 0.995 |
| FuseCodec-Fusion | Self-After | 4.07 | 6.61 | 0.93 | 3.26 | 2.95 | 3.63 | 0.995 |
| FuseCodec-Fusion | Self-Before | **3.92** | **6.36** | 0.94 | 3.43 | 3.05 | 3.59 | 0.995 |
| FuseCodec-Fusion | Cross-After | 4.17 | 6.70 | 0.93 | 3.28 | 2.90 | 3.61 | 0.995 |
| FuseCodec-Fusion | Cross-Before | 3.99 | 6.45 | **0.95** | **3.47** | **3.13** | 3.63 | 0.995 |

**Setup.** We investigate the impact of changing the attention-projection configuration in FuseCodec-Fusion (Section 2.3.1). The selected method, **Cross-Before**, applies multi-head cross-attention prior to projection:

$$\mathbf{S}' = \text{CrossAttention}(\tilde{\mathbf{S}}, \tilde{\mathbf{C}}, \tilde{\mathbf{C}})\mathbf{W}_S,$$
$$\mathbf{C}' = \text{CrossAttention}(\tilde{\mathbf{C}}, \tilde{\mathbf{S}}, \tilde{\mathbf{S}})\mathbf{W}_C, \tag{8}$$

where $\tilde{\mathbf{S}}, \tilde{\mathbf{C}} \in \mathbb{R}^{T' \times D'}$ are broadcasted global semantic and contextual vectors. We compare this with the following ablated variants:

**None**, which skips attention and directly applies projection:

$$\mathbf{S}' = \tilde{\mathbf{S}}\mathbf{W}_S,$$
$$\mathbf{C}' = \tilde{\mathbf{C}}\mathbf{W}_C \tag{9}$$

**Self-Before**, which applies self-attention before projection:

$$\mathbf{S}' = \text{SelfAttention}(\tilde{\mathbf{S}}, \tilde{\mathbf{S}}, \tilde{\mathbf{S}})\mathbf{W}_S,$$
$$\mathbf{C}' = \text{SelfAttention}(\tilde{\mathbf{C}}, \tilde{\mathbf{C}}, \tilde{\mathbf{C}})\mathbf{W}_C \tag{10}$$

**Self-After**, which projects first and then applies self-attention:

$$\mathbf{S}' = \text{SelfAttention}(\tilde{\mathbf{S}}\mathbf{W}_S),$$
$$\mathbf{C}' = \text{SelfAttention}(\tilde{\mathbf{C}}\mathbf{W}_C) \tag{11}$$

**Cross-After**, which applies projection before cross-attention:

$$\mathbf{S}' = \text{CrossAttention}(\tilde{\mathbf{S}}\mathbf{W}_S, \tilde{\mathbf{C}}\mathbf{W}_C, \tilde{\mathbf{C}}\mathbf{W}_C),$$
$$\mathbf{C}' = \text{CrossAttention}(\tilde{\mathbf{C}}\mathbf{W}_C, \tilde{\mathbf{S}}\mathbf{W}_S, \tilde{\mathbf{S}}\mathbf{W}_S) \tag{12}$$

**Results.** Table 6 shows the results of five variants. The selected **Cross-Before** setup achieves the highest performance on intelligibility STOI (0.95), and all naturalness metrics: ViSQOL (3.47), PESQ (3.13), and second-best UTMOS (3.63). **Self-Before** yields the best WER (3.92) and WIL (6.36), and second-best ViSQOL (3.43), PESQ (3.05), and STOI (0.94). The **None** and **Cross-After** configurations perform comparatively worse across intelligibility and naturalness.

**Discussion.** These results demonstrate that the configuration of attention relative to projection significantly impacts the effectiveness of representation fusion. The best-performing method, **Cross-Before**, applies cross-modal attention in the original lower-dimensional space. This enables richer semantic-contextual interactions to be captured before transformation into the higher-dimensional encoder space, leading to improved intelligibility and perceptual quality.

**Self-Before** performs competitively by achieving the best WER and WIL, suggesting that intra-modal structuring of global feature representations also benefits the fusion approach. However, the absence of explicit cross-modal exchange limits its effectiveness on naturalness metrics such as UTMOS and PESQ.

By contrast, **Cross-After** performs poorly, indicating that applying cross-attention after projection diminishes its effectiveness. Suggesting that once projected into the higher-dimensional space, the global vectors lose semantic coherence, resulting in less expressive fusion and lower audio quality.

Finally, removing attention (**None**) results in the weakest performance on intelligibility and perceptual scores, despite yielding the highest UTMOS. This indicates that even unstructured modality signals can enhance naturalness, but without alignment through attention mechanisms, they fail to deliver consistent semantic-contextual grounding.

Overall, these results confirm that performing attention prior to projection, especially cross-modal attention, is essential for extracting the most benefit from semantic-contextual signals during fusion.

### D.2 ABLATION: ATTENTION-GUIDANCE CONFIGURATION IN SEMANTIC-CONTEXTUAL GUIDANCE

**Setup.** We study the impact of attention configuration and guidance modality used in the distillation objective. Our method, FuseCodec-Distill, introduces timestep-aligned supervision using global

Table 7: Ablation of attention and guidance strategies in semantic-contextual distillation. **Cross** variants apply cross-attention between contextual embeddings and discrete tokens, while **None** applies supervision directly. **Semantic-Contextual** combines both global semantic and contextual signals. *Direct supervision using both signals achieves the best intelligibility and perceptual quality by preserving global structure.*

| Model Variant | Attention | Guidance | Content Preservation | | | Speech Naturalness | | | |
|---|---|---|---|---|---|---|---|---|---|
| | | | WER↓ | WIL↓ | STOI↑ | ViSQOL↑ | PESQ↑ | UTMOS↑ | Similarity↑ |
| FuseCodec-Distill | None | Contextual | 4.20 | 6.77 | 0.93 | 3.13 | 2.74 | 3.60 | 0.995 |
| FuseCodec-Distill | Cross | Contextual | 4.18 | 6.75 | 0.93 | 3.21 | 2.83 | 3.60 | 0.995 |
| FuseCodec-Distill | None | Semantic-Contextual | **4.09** | **6.60** | **0.94** | **3.43** | **3.06** | **3.65** | **0.996** |
| FuseCodec-Distill | Cross | Semantic-Contextual | 4.21 | 6.82 | 0.93 | 3.18 | 2.84 | 3.62 | 0.994 |

contextual and semantic signals (Section 2.3.2). The selected configuration, **None + Semantic-Contextual**, projects the first-layer RVQ tokens $\mathbf{Q}^{(1)}$ and computes cosine similarity with both semantic and contextual guidance vectors:

$$
\begin{aligned}
\mathcal{L}_{\text{distill}} = -\frac{1}{T'} \sum_{t=1}^{T'} \log \sigma \Bigg( \frac{1}{2} \Big[ \cos \Big( \mathbf{Q}_t'^{(1)}, \tilde{\mathbf{S}}_t \Big) \\
+ \cos \Big( \mathbf{Q}_t'^{(1)}, \tilde{\mathbf{C}}_t \Big) \Big] \Bigg)
\end{aligned}
\tag{13}
$$

We compare this against three ablated variants:

**None + Contextual**, which excludes both attention and semantic guidance:

$$
\mathcal{L}_{\text{distill}} = -\frac{1}{T'} \sum_{t=1}^{T'} \log \sigma \Big( \cos \Big( \mathbf{Q}_t'^{(1)}, \tilde{\mathbf{C}}_t \Big) \Big)
\tag{14}
$$

**Cross + Contextual**, which introduces cross-attention between contextual vectors and projected RVQ tokens:

$$
\tilde{\mathbf{C}} = \text{CrossAttention}(\tilde{\mathbf{C}}, \mathbf{Q}'^{(1)}, \mathbf{Q}'^{(1)})
\tag{15}
$$

**Cross + Semantic-Contextual**, which includes cross-attention but retains both guidance signals.

**Results.** Table 7 reports the performance across four configurations. The best-performing variant is **None + Semantic-Contextual**, achieving the lowest WER (4.09) and WIL (6.60), and highest scores on STOI (0.940), ViSQOL (3.43), PESQ (3.06), UTMOS (3.65), and Similarity (0.996). The second-best results are obtained by **Cross + Contextual**, but excluding semantic guidance or using attention degrades performance across all metrics.

**Discussion.** These results show that including both semantic and contextual supervision is essential for improving the quantization quality of the discrete tokens. The **None + Semantic-Contextual** configuration outperforms all others, highlighting that cosine-based alignment with both modalities provides the most stable and effective guidance during quantized representation learning.

Introducing cross-attention (**Cross**) reduces performance, suggesting that attention distorts the global nature of the guidance signals and makes supervision less consistent across time. The **Cross + Semantic-Contextual** variant also underperforms, despite having access to both guidance sources, indicating that attention interferes with their inherent structure and alignment function.

The **Contextual-only** variants perform comparatively worse, confirming that semantic signals play an important role in guiding the learned representations toward higher-level content fidelity and improved intelligibility.

Overall, these findings support using both guidance signals in their original global forms and applying them directly, without attention, to ensure stable, timestep-aligned distillation.

Table 8: Ablation of windowing and guidance strategies in temporally aligned contextual supervision. **Dynamic** variants adapt the alignment window per token based on content similarity, while **Fixed** variants use a uniform window. **Semantic-Contextual** combines semantic and contextual signals for supervision. *Dynamic windowing consistently improves intelligibility and clarity by enabling finer temporal alignment of contextual embeddings.*

| Model Variant | Window | Guidance | Content Preservation | | | Speech Naturalness | | | |
|---|---|---|---|---|---|---|---|---|---|
| | | | WER↓ | WIL↓ | STOI↑ | ViSQOL↑ | PESQ↑ | UTMOS↑ | Similarity↑ |
| FuseCodec-ContextAlign | Fixed | Contextual | 4.26 | 6.88 | 0.92 | **3.19** | 2.71 | 3.58 | 0.994 |
| FuseCodec-ContextAlign | Dynamic | Contextual | **4.15** | **6.70** | **0.93** | 3.18 | **2.85** | 3.65 | 0.995 |
| FuseCodec-ContextAlign | Fixed | Semantic-Contextual | 4.30 | 6.88 | 0.92 | 3.10 | 2.62 | 3.74 | 0.995 |
| FuseCodec-ContextAlign | Dynamic | Semantic-Contextual | 4.21 | 6.78 | 0.93 | 3.12 | 2.72 | **3.75** | 0.995 |

### D.3 Ablation: Fixed vs. Dynamic Window Configuration in Temporal Alignment

**Setup.** We investigate the effect of **fixed** versus **dynamic** windowing in the token alignment algorithm (Algorithm 1). Our full method, FuseCodec-ContextAlign, aligns each contextual embedding $\mathbf{C}_i \in \mathbb{R}^{D'}$ to a localized region of RVQ tokens $\{\mathbf{Q}_t^{(1)}\}_{t=1}^{T'}$ based on cosine similarity. The selected configuration, **Dynamic-window Contextual** (see Section 2.3.3), dynamically adjusts the alignment window for each $\mathbf{C}_i$, using the index of the previous match to guide the next search range. This content-aware strategy produces a temporally aligned sequence $\mathbf{C}^* \in \mathbb{R}^{T' \times D'}$, which is used to compute a timestep-level distillation loss:

$$\mathcal{L}_{\text{align}} = -\frac{1}{T'} \sum_{t=1}^{T'} \log \sigma \left( \cos \left( \mathbf{Q}_t'^{(1)}, \mathbf{C}_t^* \right) \right) \tag{16}$$

We compare this setup against the following ablated variants:

**Fixed-window Contextual**, which uses a fixed alignment window of size $w = \lfloor T'/n \rfloor$, where $T'$ is the RVQ sequence length and $n$ is the number of contextual embeddings. Each $\mathbf{C}_i$ is aligned to the most similar token $\mathbf{Q}_t^{(1)}$ within its predefined window.

**Fixed-window Semantic-Contextual**, which adds semantic supervision using semantic representations $\{\mathbf{S}_i\}_{i=1}^m$, in addition to contextual representations aligned via a fixed-window token alignment. Since both semantic and RVQ tokens are extracted at the same frame rate, they are inherently time-aligned, requiring no additional alignment. The combined loss is:

$$\mathcal{L}_{\text{align}} = -\frac{1}{T'} \sum_{t=1}^{T'} \log \sigma \left( \frac{1}{2} \left[ \cos \left( \mathbf{Q}_t'^{(1)}, \mathbf{C}_t^* \right) \right. \right.$$
$$\left. \left. + \cos \left( \mathbf{Q}_t'^{(1)}, \mathbf{S}_t \right) \right] \right) \tag{17}$$

**Dynamic-window Semantic-Contextual**, which replaces the fixed window with a dynamic alignment strategy, while also incorporating direct supervision from semantic embeddings $\{\mathbf{S}_t\}$.

**Results.** As shown in Table 8, the **Dynamic-window Contextual** configuration achieves the best performance across content preservation metrics, achieving the lowest WER (4.15), WIL (6.70), and highest STOI (0.93). It also performs strongly in terms of speech naturalness, with the best PESQ (2.85), high ViSQOL (3.18), and top Similarity (0.995). The **Dynamic Semantic-Contextual** variant achieves the best UTMOS (3.75), second-best WER (4.21) and WIL (6.78), and matches the top Similarity. By contrast, both **Fixed-window** configurations obtains lower scores across most metrics, particularly the **Fixed Semantic-Contextual** configuration, which scores the lowest ViSQOL (3.10) and PESQ (2.62), despite a relatively high UTMOS (3.74).

**Discussion.** These results highlight the importance of the temporal alignment strategy in influencing speech reconstruction quality. The superior performance of the **Dynamic-window Contextual** variant demonstrates that token alignment using a dynamic window, where contextual embeddings

are adaptively aligned based on token similarity, achieves better semantic grounding and contextual precision.

In contrast, the **Fixed-window** variants suffer from rigid alignment constraints. They fail to capture fine-grained temporal dependencies by enforcing a fixed windowing strategy, which resuls in degraded speech clarity (lower ViSQOL and PESQ). This limitation is especially noticeable in the **Fixed Semantic-Contextual** setup, where the addition of semantic supervision is insufficient to compensate for the strictly aligned contextual embeddings as the fixed window does not account for local content variations.

Both **Semantic-Contextual** variants improve UTMOS, indicating that semantic supervision contributes positively to speech naturalness. However, this comes with a trade-off when not paired with dynamically aligned contextual guidance, as the semantic-only supervision fails to improve content accuracy.

Overall, these findings underscore that dynamic alignment is essential for effective contextual representation guidance. They also highlight that while semantic supervision enhances fluency and naturalness, it must be combined with flexible alignment mechanisms to avoid compromising content preservation.

### D.4 ABLATION: DROPOUT MASK CONFIGURATION IN REPRESENTATION FUSION

Table 9: Ablation of modality dropout probability during latent representation fusion in FuseCodec. **Dropout** indicates the stochastic masking rate applied independently to semantic and contextual representations during training. Moderate dropout prevents over-reliance on a single modality, while higher rates degrade multimodal integration. *A 10% dropout rate achieves the best trade-off, maximizing intelligibility and perceptual quality.*

| Model Variant | Dropout | Content Preservation | | | Speech Naturalness | | | |
|---|---|---|---|---|---|---|---|---|
| | | WER↓ | WIL↓ | STOI↑ | ViSQOL↑ | PESQ↑ | UTMOS↑ | Similarity↑ |
| FuseCodec-Fusion | 10% | **3.99** | **6.45** | **0.95** | **3.47** | **3.13** | 3.63 | 0.995 |
| FuseCodec-Fusion | 30% | 4.10 | 6.63 | 0.94 | 3.29 | 2.96 | 3.65 | 0.995 |
| FuseCodec-Fusion | 50% | 4.09 | 6.58 | 0.94 | 3.33 | 2.97 | **3.66** | **0.996** |
| FuseCodec-Fusion | 70% | 4.08 | 6.64 | 0.93 | 3.26 | 2.91 | 3.63 | 0.995 |
| FuseCodec-Fusion | 90% | 4.15 | 6.67 | 0.93 | 3.26 | 2.86 | 3.61 | 0.995 |

**Setup.** We investigate the effect of modality dropout rate on the quality of latent representation fusion. As described in Section 2.3.1, we apply stochastic dropout masks $\mathcal{D}_S, \mathcal{D}_C \in \{0,1\}^{T' \times D}$ element-wise to the projected semantic ($\mathbf{S}'$) and contextual ($\mathbf{C}'$) vectors during training:

$$\mathbf{Z}' = \mathbf{Z} + (\mathbf{S}' \odot \mathcal{D}_S) + (\mathbf{C}' \odot \mathcal{D}_C) \tag{18}$$

This stochastic masking prevents FuseCodec from over-reliance on any single modality and encourages the model to learn robust representations.

The selected configuration uses a **10%** dropout rate—i.e., each element in $\mathcal{D}_S$ and $\mathcal{D}_C$ has a 10% chance of being masked to zero during training. We compare this against higher dropout rates: **30%**, **50%**, **70%**, and **90%**.

**Results.** The best overall performance is achieved with the 10% dropout rate configuration, which achieves the lowest WER (3.99) and WIL (6.45) and the highest STOI (0.95), ViSQOL (3.47), and PESQ (3.13). Increasing the dropout rate to 30–90% leads to the worsening of the most content preservation and speech naturalness metrics. While UTMOS and Similarity remain relatively stable, 50% dropout achieves minor gains in UTMOS (3.66) and Similarity (0.996).

**Discussion.** These results confirm the importance of carefully balancing modality dropout during latent fusion and underscore the value of semantic-contextual representation integration. Preserving a sufficient portion of the auxiliary representations by using a small 10% dropout rate achieves the most effective use of semantic and contextual information.

As the dropout rate increases, the model receives increasingly less additional modality information, reducing its ability to align latent tokens with multimodal supervision. This negatively affects intelligibility (WER, WIL) and perceptual quality (ViSQOL, PESQ).

Interestingly, metrics such as UTMOS and Similarity remain relatively stable or improve at moderate dropout rates (50%), suggesting that prosodic and speaker characteristics are preserved within the base latent representations. However, the loss of some semantic-contextual information comes at the cost of worse content preservation.

Overall, the findings suggest that light dropout (10%) provides the best trade-off, ensuring robust yet expressive multimodal grounding during latent token fusion.

### D.5 ABLATION: QUNTIZER LAYER CONFIGURATION IN SEMANTIC-CONTEXTUAL GUIDANCE

Table 10: Ablation of RVQ supervision depth under global (Distill) and temporally aligned (ContexAlign) guidance. **First Layer** indicates supervision is applied only to the first-layer RVQ tokens, while **All Layers** averages representations from all eight RVQ layers before supervision. *Supervising the first-layer RVQ tokens leads to stronger semantic-contextual grounding and improved intelligibility compared to all-layer supervision.*

| Model Variant | RVQ Layer | Content Preservation | | | Speech Naturalness | | | |
|---|---|---|---|---|---|---|---|---|
| | | WER↓ | WIL↓ | STOI↑ | ViSQOL↑ | PESQ↑ | UTMOS↑ | Similarity↑ |
| FuseCodec-ContextAlign | First Layer | **4.15** | **6.70** | 0.93 | **3.18** | **2.85** | **3.65** | **0.995** |
| FuseCodec-ContextAlign | All Layers | 4.34 | 7.04 | 0.93 | 3.17 | 2.72 | 3.65 | 0.993 |
| FuseCodec-Distill | First Layer | **4.09** | **6.60** | **0.94** | **3.43** | **3.06** | **3.65** | **0.996** |
| FuseCodec-Distill | All Layers | 4.23 | 6.86 | 0.93 | 3.26 | 2.84 | 3.61 | 0.994 |

**Setup.** We study the impact of RVQ layer supervision depth in the distillation objective. Our method, FuseCodec-Distill, uses **first-layer supervision**, projecting the first-layer RVQ tokens $\mathbf{Q}^{(1)}$ and computing cosine similarity (see Sections 2.3.2 and 2.3.3).

We compare this against an ablated variant, **all-layer supervision**, which averages the outputs from all eight RVQ layers. We define the averaged RVQ output as:

$$
\mathbf{Q}^{(1:8)} = \frac{1}{8} \sum_{i=1}^{8} \mathbf{Q}^{(i)} \in \mathbb{R}^{T' \times D},
$$

$$
\mathbf{Q}'^{(1:8)} = \mathbf{Q}^{(1:8)} \mathbf{W}
$$

(19)

In the **Global Semantic-Contextual Supervision** setting, we apply the **all-layer supervision** to the distillation loss as:

$$
\mathcal{L}_{\text{distill}} = -\frac{1}{T'} \sum_{t=1}^{T'} \log \sigma \left( \frac{1}{2} \left[ \cos \left( \mathbf{Q}'^{(1:8)}_t, \tilde{\mathbf{S}}_t \right) \right. \right.
$$
$$
\left. \left. + \cos \left( \mathbf{Q}'^{(1:8)}_t, \tilde{\mathbf{C}}_t \right) \right] \right)
$$

(20)

Similarly, for the **Temporally Aligned Contextual Supervision** setting, we apply the **all-layer supervision** to the distillation loss as:

$$
\mathcal{L}_{\text{align}} = -\frac{1}{T'} \sum_{t=1}^{T'} \log \sigma \left( \cos \left( \mathbf{Q}'^{(1:8)}_t, \mathbf{C}^*_t \right) \right)
$$

(21)

**Results.** Table 10 shows the effect of RVQ supervision depth across both distillation configurations. For FuseCodec (Distill), which uses Global Semantic-Contextual Supervision, first-layer supervision achieves the strongest performance across all content preservation and naturalness metrics, with the lowest WER (4.09), WIL (6.60), and highest STOI (0.94), ViSQOL (3.43), PESQ (3.06), UTMOS (3.65), and Similarity (0.996). Similarly, FuseCodec (ContexAlign), which uses Temporally Aligned Contextual Supervision, First-layer supervision again achieves stronger results in WER (4.15), WIL (6.70), ViSQOL (3.18), PESQ (2.85), and Similarity (0.995). In contrast, using all-layer supervision leads to consistent degradation across most metrics in both settings.

**Discussion.** The results highlight that the layer at which RVQ tokens are supervised significantly impacts the quality of semantic and contextual guidance during distillation. Supervising the first RVQ layer yields stronger performance, as these tokens encode high-level, abstract representations more aligned with semantic intent and global context. This leads to better linguistic grounding and intelligibility, reflected in improved WER, STOI, and ViSQOL scores.

In contrast, deeper RVQ layers capture lower-level acoustic and residual details, which are less suitable for semantic or contextual alignment. Averaging supervision across all layers matches these fine-grained signals with global ones, impacting the alignment objective. This results in performance drop across content preservation and speech naturalness metrics.

Some naturalness metrics, such as UTMOS and Similarity, remain relatively stable with all-layer supervision, suggesting that speaker identity and prosodic features are distributed throughout the RVQ layers. However, these are insufficient for guiding semantic alignment during distillation.

Overall, applying supervision at the first RVQ layer provides a clearer, more semantically grounded signal, leading to better alignment and overall performance in speech reconstruction.

# E   Tokenizer Design and Loss Functions

In this section, we provide additional details on our tokenizer backbone (§E.1) and the training objectives for the backbone neural codec (§E.2).

## E.1   Model Details

To implement a strong speech tokenizer baseline, we adopt a standard neural codec architecture and discriminator setup commonly used in prior work Défossez et al. (2022); Zeghidour et al. (2022).

**Encoder and Decoder.** The Encoder consists of an initial 1D convolutional layer with 32 channels and a kernel size of 7, followed by 4 stacked residual blocks. Each block includes two dilated convolutions with a (3, 1) kernel and no dilation expansion (dilation = 1), a residual connection, and a strided convolutional layer for temporal downsampling. Stride values across the blocks are set to 2, 4, 5, and 8, with kernel sizes for the downsampling layers set to twice the corresponding stride. Channel dimensions double at each downsampling stage. The encoder then includes a two-layer BiLSTM, and concludes with a 1D convolution (kernel size 7) to project to the target embedding dimension. ELU (Clevert et al., 2016) is used as the activation function, and layer normalization or weight normalization is applied depending on the layer. The Decoder mirrors the encoder architecture, with the only difference being the use of transposed convolutions in place of strided convolutions to reverse the downsampling steps, and the inclusion of LSTM layers to restore temporal resolution.

**Residual Vector Quantizer.** The Residual Vector Quantizer (RVQ) module discretizes the encoder's continuous latent representations into a sequence of codebook indices. Specifically, we quantize the encoder latent tensor of shape $[B, D, T]$ using 8 residual codebooks, each with 1024 codebook entries. Each subsequent codebook quantizes the residual error of the previous one. Codebook entries are updated using an exponential moving average with a decay factor of 0.99. To prevent codebook collapse, unused entries are randomly resampled using vectors from the current batch. The RVQ output is a discrete tensor of shape $[B, N_q, T]$, where $N_q$ is the number of active quantizers. The indices are mapped back to the original latent space by summing the corresponding codebook embeddings and are then fed into the decoder to reconstruct the input. A straight-through estimator (Bengio et al., 2013) is used to propagate gradients through the quantizer.

**Discriminators.** We utilize discriminators to guide the generators (Encoder, RVQ, and Decoder) to reconstruct speech more closely to the original. We make use of three distinct discriminators: a Multi-Scale STFT (MS-STFT) discriminator, a Multi-Scale Discriminator (MSD), and a Multi-Period Discriminator (MPD). The MS-STFT discriminator, proposed by (Défossez et al., 2022), works on multiple resolutions of the complex-valued short-time Fourier transform (STFT). It treats the real and imaginary parts as concatenated and applies a sequence of 2D convolutional layers. The initial layer uses a kernel size of $3 \times 8$ with 32 channels. This is followed by convolutions with increasing temporal dilation rates (1, 2, and 4) and a stride of 2 along the frequency axis. A final $3 \times 3$ convolution with stride 1 outputs the discriminator prediction. The MSD processes the

raw waveform at various temporal scales using progressively downsampled versions of the input. We adopt the configuration from (Zeghidour et al., 2022), which was originally based on (Kumar et al., 2019). Similarly, the MPD, introduced by (Kong et al., 2020), models periodic structure in the waveform by reshaping it into a 2D input with unique periodic patterns. For consistency, we standardize the number of channels in both the MSD and MPD to match those in the MS-STFT discriminator.

### E.2 TRAINING OBJECTIVE

To ensure that FuseCodec learns discrete speech representations, we ground our training objective on proven techniques, following (Défossez et al., 2022; Zhang et al., 2024; Ahasan et al., 2024).

**Reconstruction loss.** Let $\mathbf{x}$ and $\hat{\mathbf{x}}$ denote the original and reconstructed speech waveforms, respectively. For spectral comparisons, we define 64-bin Mel-spectrograms $\mathbf{M}_i(\cdot)$ using STFTs with window size $2^i$ and hop size $2^i/4$, where $i \in \mathcal{E} = \{5, \dots, 11\}$ indexes different resolution scales. We compute the time-domain $\mathcal{L}_{\text{time}}$ and frequency-domain $\mathcal{L}_{\text{freq}}$ reconstruction losses as:

$$\mathcal{L}_{\text{time}} = \|\mathbf{x} - \hat{\mathbf{x}}\|_1 \tag{22}$$

$$\mathcal{L}_{\text{freq}} = \sum_{i \in \mathcal{E}} \bigg( \|\mathbf{M}_i(\mathbf{x}) - \mathbf{M}_i(\hat{\mathbf{x}})\|_1$$
$$+ \|\mathbf{M}_i(\mathbf{x}) - \mathbf{M}_i(\hat{\mathbf{x}})\|_2 \bigg) \tag{23}$$

**Adversarial loss.** To reduce the discriminability of reconstructed speech, we adopt a GAN-based training objective with a set of discriminators $\{D^{(i)}\}_{i=1}^{d}$, including multi-period (MPD), multi-scale (MSD), and multi-scale STFT (MS-STFT) variants (see Sec. E for details). The generator $\mathcal{L}_{\text{gen}}$ and discriminator $\mathcal{L}_{\text{disc}}$ losses are computed as:

$$\mathcal{L}_{\text{gen}} = \frac{1}{d} \sum_{i=1}^{d} \max\left(0,\ 1 - D^{(i)}(\hat{\mathbf{x}})\right) \tag{24}$$

$$\mathcal{L}_{\text{disc}} = \frac{1}{d} \sum_{i=1}^{d} \big[ \max(0,\ 1 - D^{(i)}(\mathbf{x}))$$
$$+ \max(0,\ 1 + D^{(i)}(\hat{\mathbf{x}}))\big] \tag{25}$$

Let $D_j^{(i)}(\cdot)$ denote the output of the $j$-th layer of $D^{(i)}$, with $\ell$ total layers. We include a feature $\mathcal{L}_{\text{feat}}$ matching loss to stabilize training and align intermediate features as:

$$\mathcal{L}_{\text{feat}} = \frac{1}{d\ell} \sum_{i=1}^{d} \sum_{j=1}^{\ell} \frac{\|D_j^{(i)}(\mathbf{x}) - D_j^{(i)}(\hat{\mathbf{x}})\|_1}{\text{mean}\left(\|D_j^{(i)}(\mathbf{x})\|_1\right)} \tag{26}$$

**Commitment Loss.** To ensure encoder outputs align closely with their quantized representations, we apply a commitment penalty during residual vector quantization (RVQ). Let $\mathbf{r}_j$ denote the residual vector at step $j \in \{1, \dots, q\}$, and $\mathbf{c}_j$ be its corresponding nearest codebook entry, we calculate commitment loss $\mathcal{L}_{\text{commit}}$ as:

$$\mathcal{L}_{\text{commit}} = \sum_{j=1}^{q} \|\mathbf{r}_j - \mathbf{c}_j\|_2^2 \tag{27}$$

# F QUALITATIVE COMPARISON

Figure 2 compares qualitative speech reconstruction results of FuseCodec with several baseline models, including SpeechTokenizer, DM-Codec, and EnCodec. Each row corresponds to a model, and each column shows a distinct speech sample; clicking an image opens the corresponding audio.

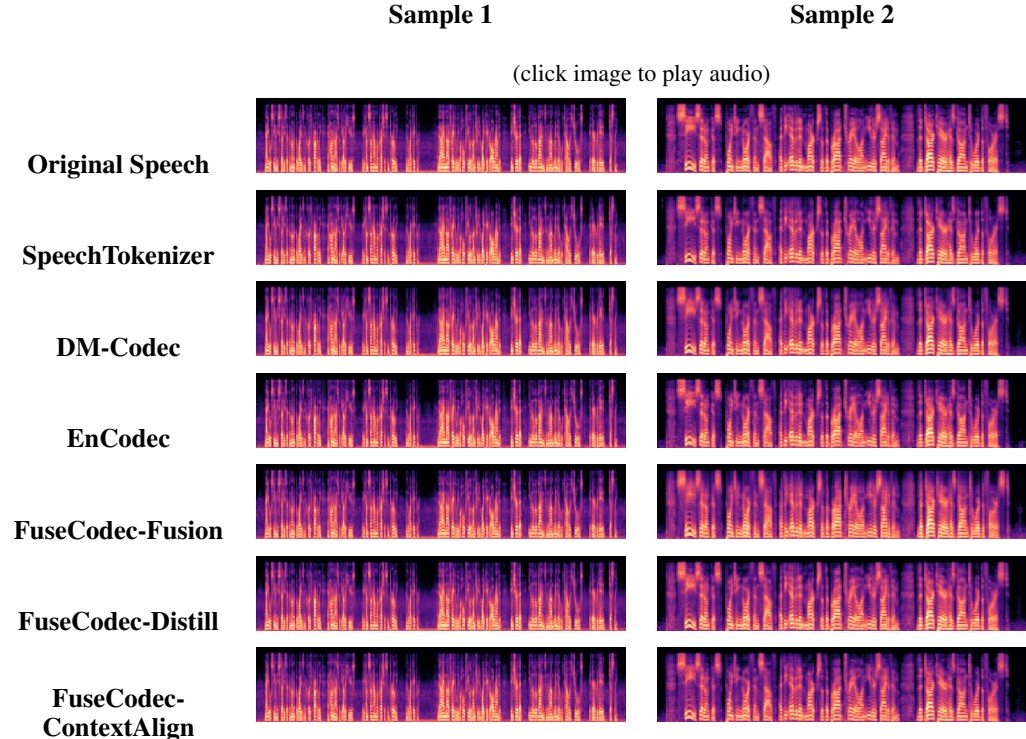

Figure 2: Qualitative speech reconstruction results for FuseCodec and baseline models. Each cell shows the spectrogram for two samples; clicking an image links to the corresponding audio.

