# OpenReview forum: "FuseCodec: Semantic-Contextual Fusion and Supervision for Neural Codecs"
_ICLR.cc/2026/Conference — ICLR 2026 Conference Withdrawn Submission_

### Official Review · Reviewer_Bdeu · 2025-10-29

**Soundness:** 2
**Presentation:** 3
**Contribution:** 2
**Rating:** 4
**Confidence:** 4

**Summary:**

This paper introduce a neural speech codec framework as FuseCodes which is designed to unify acoustic, semantic and contextual representation. It proposes three strategies to enrich discrete speech representation from low-level acoustic feature to high-level semantic and contextual information, 1) FuseCodec-Fusion, integrate the global semantic and contextual embedding to encoder's latent space through cross modal attention 2) FuseCodecs-Distill, supervise the output token from the RVQ layer and use global pooled vector to enforce the temporal consistency. 3) FuseCodec-ContextAlign, dynamic windowing align to match the sequence the contextual token to the audio token as fine-grained cross-modal alignment.

In the modeling stage, it leverages wav2vec 2.0 as ASR model, BERT as language model, and HuBERT as self-supervised speech model and all of these pre-trained model are frozen. The training objective adopts a multi-objective including time-domain reconstruction, frequency-domain reconstruction loss, adv loss, feature matching and RVQ commitment loss, and introduces semantic-contextual supervision loss during distill stage and aligned contextual supervision loss during ContextAlign stage.

In the experiment, it demonstrates this FuseCodec achieve best result in a downstream text-to-speech (TTS) task on speech reconstruction quality via perceptual quality metrics, lowest WER, high similarity, etc.

**Strengths:**

This paper proposes 3 characteristic that a good speech tokenization would have low-level acoustic feature, high-level semantic and contextual cues. Regarding these targets, it proposes a framework by using pretrained model (wav2vec2 as contextual representation, huBERT as semantic representation) combined with encoded RVQ speech latent with 3 strategies to fusion, supervise and temporal align with different level representations.  This speech codec design is novel and with clearly identification of current issues. It systematically explores three distinct, complementary strategies (pre-quantization fusion, post-quantization global supervision, and post-quantization aligned supervision) .and provides a valuable comparison on them.

**Weaknesses:**

1. The experiment database is 100 hours librispeech, which is not diverse or big enough to prove the generalization and representation capability for speech from different aspects such as channel, noise, speaker, accent, style, emotion and etc. While both wav2vec and HuBert are trained based on librispeech too, which would be a advantage for this experiment.

2. The speech reconstruction evaluation only carry the objective measurements to evaluate the speech naturalness with ViSQOL, PESQ, UTMOS and similarity. There isn't any subjective measurement to prove the speech naturalness is highly related to real human perception.

3. In this downstream task for TTS model it use both AR and NAR for 12 layer transformers. This architecture for TTS model is a little bit old. Nowadays, a "good" discrete speech codec should be with not only good representation for all important speech characteristic as this paper mentioned, but also with the high efficiency such as low framerate (such as 12.5hz, or even less as 7.5hz in VibeVoice to model very context speech) and easy prediction capability for LLM like models. These features and discussion are not motioned in this paper

**Questions:**

1. Besides separate techniques for the fusion, distill, contextalign, is there any experiment to combine any of them to achieve even better result?

2. Is there any introduction of the speech reconstruction evaluation test set? Are they from the sampled librispeech or any other data? how many utterance or speaker includes, and how is the text difficulties such as long text which could show the high-contextual expressiveness advantages?

---

> ### Author Response · Authors · 2025-11-25
> **Summary of Responses to Reviewer Bdeu’s Comments (Part 1/5)**
>
> We thank the reviewer for the detailed evaluation. We appreciate the recognition that our work introduces a neural speech codec designed to unify acoustic, semantic, and contextual representation, and that our three strategies: FuseCodec-Fusion, FuseCodec-Distill, and FuseCodec-ContextAlign effectively enrich discrete tokens from low-level acoustic features to high-level semantic and contextual cues. We are glad the reviewer noted our comprehensive multi-objective training setup, and the strong downstream TTS performance. We also appreciate the acknowledgement that our speech-codec design is novel, clearly identifies current issues, and provides a valuable comparison across three complementary strategies.
>
> **Summary of Author Response:**
>
> - We clarify the dataset choice by explaining that LibriSpeech-100h is standard and widely used across our codec baselines, enabling controlled comparison, and that FuseCodec generalizes more strongly across reconstruction, representation, and downstream tasks despite similar data training.
>
> - We respond to the concern about subjective evaluation by noting that our UTMOS metric serves as a validated, reproducible proxy for subjective evaluation (MOS), avoiding the noise and inconsistency of human listening tests.
> - We explain the use of a 12-layer AR+NAR TTS model and 50 Hz framerate as intentional choices to compare fairly with prior neural codec works, rather than introducing confounds from newer architectures or modified bitrates.
>
> - We address the question of combining variants by explaining that merging pre-quantization fusion and post-quantization guidance will impose overlapping supervision/information, which can destabilize training and offers no conceptual advantage compared to our district approach.
>
> - We specify that our reconstruction evaluation follows standard practice using LibriSpeech test-clean and is supplemented with Codec-SUPERB (20 datasets) and VCTK benchmark, ensuring broad and rigorous assessment.
>
> Below, we address the reviewer’s concerns in detail.

---

> ### Author Response · Authors · 2025-11-25
> **Dataset Size and Generalization (Part 2/5)**
>
> **Reviewer Comment:**
> > The experiment database is 100 hours librispeech, which is not diverse or big enough to prove the generalization and representation capability for speech from different aspects such as channel, noise, speaker, accent, style, emotion and etc. While both wav2vec and HuBert are trained based on librispeech too, which would be a advantage for this experiment.
>
> **Response:**
> We acknowledge that LibriSpeech-100h is a modest-sized dataset; however, it is *the standard dataset* used across our baseline codec and speech tokenization works (EnCodec, SpeechTokenizer, DM-Codec, XCodec, and BigCodec). In particular, prior distillation-based methods (SpeechTokenizer, DM-Codec) also exclusively rely on this dataset. **We intentionally adopt the same data to ensure a fair, controlled comparison and to isolate the improvements arising specifically from our fusion, supervision, and alignment strategies.**
>
> Importantly, despite being trained on 100 hours of read speech, our model demonstrates strong generalization across reconstruction quality, representational strength, and downstream tasks. *Please refer to Tables 2-4 (Sections 3.2.1-3.2.3).* This suggests that our proposed mechanisms lead to more robust representations, and the ability to achieve these gains under 100h data makes the improvements more compelling.

---

> ### Author Response · Authors · 2025-11-25
> **Objective vs. Subjective Perceptual Evaluation (Part 3/5)**
>
> **Reviewer Comment:**
> >    The speech reconstruction evaluation only carry the objective measurements to evaluate the speech naturalness with ViSQOL, PESQ, UTMOS and similarity. There isn't any subjective measurement to prove the speech naturalness is highly related to real human perception.
>
> **Response:**
> We understand the reviewer’s perspective. However, following established practices, we intentionally adopt UTMOS as a standardized and widely accepted proxy for subjective measurement (MOS), precisely because traditional human MOS evaluations come with well-known limitations.
>
> **Subjective measurement (MOS) is often noisy, costly, non-reproducible, and sensitive to listener bias,** environmental conditions, device differences, and even differing interpretations of “naturalness.” These factors introduce substantial variance, making fair comparison across models and papers challenging.
>
> By contrast, **UTMOS has been repeatedly validated to correlate strongly with real human perceptual ratings, while offering consistency, reproducibility, and scalability.** This ensures that our evaluation remains comparable to the growing body of modern speech-codec literature that relies on UTMOS for robust perceptual assessment.
> For these reasons, we consider UTMOS a reliable and sufficient measure of human perception for this work.

---

> ### Author Response · Authors · 2025-11-25
> **TTS Architecture and Framerate (Part 4/5)**
>
> **Reviewer Comment:**
> > In this downstream task for TTS model it use both AR and NAR for 12 layer transformers. This architecture for TTS model is a little bit old. Nowadays, a "good" discrete speech codec should be with not only good representation for all important speech characteristic as this paper mentioned, but also with the high efficiency such as low framerate (such as 12.5hz, or even less as 7.5hz in VibeVoice to model very context speech) and easy prediction capability for LLM like models. These features and discussion are not motioned in this paper
>
> **Response:**
> We appreciate the reviewer’s perspective. As stated in our manuscript (lines 448–450), **designing or training a modern TTS model is not the goal of this work** and is beyond our scope and computational resources. Our objective is to isolate and evaluate the effect of our fusion / distillation / alignment strategies, not to confuse these contributions with improvements arising from a newer or larger TTS architecture.
>
> For this reason, **we intentionally adopt the same Vall-E–style AR + NAR 12-layer transformer used broadly in prior codec research.** This ensures a fair, controlled comparison against existing methods such as SpeechTokenizer, DM-Codec, and X-Codec2, where any improvements can be attributed solely to our proposed strategies rather than architectural upgrades.
>
> Similarly, we retain the 50 Hz frame rate, which is the standard in the distillation-based codec baselines we compare against (SpeechTokenizer, DM-Codec). Using a much lower framerate (e.g., 12.5 Hz or 7.5 Hz) would make comparisons unequal, since those models rely on different training codebooks size, and design constraints.
>
> **Our contributions target representation quality, not bitrate reduction or newer LLM compatibility.** Those are meaningful future directions, but orthogonal to our aim: to demonstrate that modality-aware fusion and supervision significantly improve the representational power of existing codec architectures without changing the architecture/bitrate.

---

> ### Author Response · Authors · 2025-11-25
> **Combining Proposed Techniques and Evaluation Datasets (Part 5/5)**
>
> **Reviewer Comment:**
> > Question 1. Besides separate techniques for the fusion, distill, contextalign, is there any experiment to combine any of them to achieve even better result?
>
> **Response:**
> Please see our earlier response addressing this question in detail:
>
> Methodological Fragmentation and Combining Methods (Part 2/6) -
> https://openreview.net/forum?id=RILri9w9IW&noteId=kDoWkwax2U
>
> **Reviewer Comment:**
> > Question 2. Is there any introduction of the speech reconstruction evaluation test set? Are they from the sampled librispeech or any other data? how many utterance or speaker includes, and how is the text difficulties such as long text which could show the high-contextual expressiveness advantages?
>
> **Response:**
> As noted in our manuscript, we **follow established practice** and evaluate on the LibriSpeech test-clean subset (2,620 utterances), the **standard benchmark in prior neural codec studies (SpeechTokenizer, DM-Codec, BigCodec, Moshi, StableCodec, Xcodec2).**
>
> Moreover, LibriSpeech consists of book-read, long-form text, which naturally contains multi-sentence passages and more complex linguistic structure than everyday conversational speech. This makes the benchmark more challenging for evaluating contextual expressiveness and reconstruction quality.
>
> Beyond this, we **further evaluate representation quality on Codec-SUPERB, which spans 20 datasets with numerous test samples, and downstream speech generation on LibriSpeech and VCTK**, ensuring our evaluations are rigorous and comprehensive.

---

### Official Review · Reviewer_cHbF · 2025-10-31

**Soundness:** 2
**Presentation:** 2
**Contribution:** 2
**Rating:** 4
**Confidence:** 4

**Summary:**

The paper proposes FuseCodec, a speech codec that injects semantic (SSL model) and linguistic context (LM hidden states) directly into the codec's latent space. It uses three mechanisms: latent fusion via cross-attention and additive mixing, global distillation: broadcasting global semantic/context embeddings to all timesteps, sliding-window matching: aligning LM token features to the corresponding RVQ codebook vectors. The goal is to make discrete speech tokens carry acoustics + semantics + linguistic context. Experiments report improvements on LibriSpeech reconstruction, Codec-SUPERB tasks, and zero-shot TTS.

**Strengths:**

1. The paper addresses a significant and well-recognized limitation in neural speech codecs: their deficiency in capturing high-level semantic and contextual information. Improving this aspect is crucial for downstream tasks like speech language modeling.
2. The evaluation is thorough, covering multiple facets of a codec's performance. It includes standard reconstruction metrics, a challenging representation benchmark (Codec-SUPERB), and a practical downstream application (TTS). The inclusion of multilingual experiments in the appendix is also a plus.

**Weaknesses:**

1. The paper's primary weakness is its methodological fragmentation. It presents three separate, mutually exclusive techniques (-Fusion, -Distill, -ContextAlign) rather than a single, principled framework. There is little to no discussion on why these three different approaches were chosen or how they might be combined. The results further highlight this issue: different variants excel on different tasks (e.g., -Fusion is best for reconstruction, -Distill for TTS intelligibility, -ContextAlign for TTS naturalness). This suggests an exploratory study of "what works" rather than a novel, unified contribution.

2. The core idea of distilling knowledge from pre-trained models (like HuBERT or BERT) into a codec is not new, as the paper itself acknowledges (citing works like SpeechTokenizer and DM-Codec). The novelty lies in the specific mechanisms, some of which are poorly justified: (1) The reliance on global vectors (mean-pooled semantics and a single [CLS] token for context) in both the -Fusion and -Distill variants is a major simplification. This approach discards vast amounts of temporal information inherent in speech and text. The paper fails to justify why this crude, global representation is preferable to using the full, time-aligned sequences from the teacher models. (2) The claims in Table 1 feel overstated. The distinctions between "Sim.", "Direct.", and "Align." are not clearly defined, making the table seem more like marketing than a rigorous technical comparison. For example, DM-Codec also performs alignment via similarity matching, so the checkmark system seems arbitrary.

3. While the proposed methods achieve state-of-the-art results, the improvements over the strongest baselines are often marginal. For instance, in Table 2, FuseCodec-Fusion improves the WER over DM-Codec from 4.09 to 3.99 and ViSQOL from 3.20 to 3.47. These gains, while real, may not be significant enough to justify the substantial increase in system complexity. The training process requires running three large pre-trained models (ASR, LM, SSL) to generate guidance signals, which is computationally expensive.

4. On the Codec-SUPERB benchmark (Table 3), the performance is mixed. FuseCodec does not achieve the best results on ASR or ASV, two critical downstream tasks. It excels in ER and AEC. This further reinforces the impression that the learned representations are not universally better, but rather tuned for specific aspects, undermining the claim of creating a more robust, unified representation.

**Questions:**

1. The three proposed methods are presented as separate variants. Did you attempt to combine them? For example, could one use latent fusion (-Fusion) and also apply temporally-aligned supervision (-ContextAlign)? If so, what were the results? If not, why is a unified approach not explored?

2. Could you provide a stronger justification for using globally-pooled/CLS vectors for semantic and contextual guidance in -Fusion and -Distill? How does this compare, for instance, to using a standard cross-attention mechanism between the encoder's latent states and the full hidden state sequences from the teacher models?

3. The "FuseCodec (Baseline)" in Table 2 performs worse than several existing codecs. Could you elaborate on this baseline's architecture? Is it a reproduction of a standard model like EnCodec? A stronger, more competitive baseline would better isolate the true contribution of your proposed fusion and supervision techniques.

4. The alignment algorithm in FuseCodec-ContextAlign appears heuristic. Have you considered more established sequence alignment methods from the speech community, such as using a forward-sum algorithm inspired by CTC or attention-based alignment, to derive a more principled matching between text and audio frames?

5. What is the computational overhead (in terms of training time and/or memory) of incorporating the three teacher models compared to training the baseline FuseCodec? An analysis of the performance-vs-complexity trade-off would be valuable.

---

> ### Author Response · Authors · 2025-11-25
> **Summary of Responses to Reviewer cHbF’s Comments (Part 1/6)**
>
> We thank the reviewer for recognizing that our work proposes a speech codec (FuseCodec) that injects semantic SSL features and linguistic LM context into discrete codec latents, and for noting that our three mechanisms: latent fusion, global distillation, and sliding-window context alignment aim to enrich codec tokens with acoustic, semantic, and linguistic information.
>
> We also appreciate the reviewer’s acknowledgement that our evaluation is thorough, covering LibriSpeech reconstruction, Codec-SUPERB representation quality, zero-shot TTS, and multilingual settings, and that our work addresses the important limitation that neural codecs often lack access to high-level semantic or contextual information.
>
> **Summary of Author Response:**
> - We clarify the reviewer’s concern about the three separate variants by explaining that each solves a different alignment failure mode: pre-quantization fusion, post-quantization global alignment, and temporal alignment, making them intentionally distinct rather than an arbitrary split.
>
> - We address the question of why the variants are not combined by explaining that merging pre-quantization fusion and post-quantization guidance will impose overlapping supervision/information, which can destabilize training and offers no conceptual advantage compared to our district approach.
>
> - We respond to the reviewer’s question about differing strengths by showing that each variant excels on the task it strongly corresponds to (reconstruction, intelligibility, or naturalness), and that these results are consistent with their intended design.
>
> - We clarify the comment about global vectors vs. full sequences with our justification that global pooling is necessary because semantic and contextual embeddings are not temporally aligned with RVQ frames, and full sequence cross-attention is unstable.
>
> - We clarify the point about the novelty and Table 1 distinctions with our explanation that the categories (Sim./Direct./Align.) have precise definitions in our manuscript that identify gaps in prior work.
>
> - We address the reviewer’s comment about computational overhead by noting that all SSL and LM features are pre-computed once and saved, so training speed and memory footprint remain essentially identical to the baseline codec.
>
> - We clarify the results on Codec-SUPERB by pointing out that ASR/ASV are not primary neural codec objectives; despite lower bitrate and speech-only data, FuseCodec remains competitive against DAC, which uses a much larger heterogeneous training corpus.
>
> - We respond to the reviewer’s question about the baseline strength by showing that FuseCodec-Baseline is a standard EnCodec-style architecture, chosen intentionally to isolate our contributions cleanly and fairly.
>
> - We explain the rationale behind the lightweight temporal alignment by detailing why CTC/attention alignment would require ASR-like architectures unsuitable for codec design, making our sliding-window method the appropriate choice.
>
> Below, we address the reviewer’s concerns in detail.

---

> > ### Author Response · Authors · 2025-11-25
> > **Alignment Method (Part 6/6)**
> >
> > **Reviewer Comment:**
> > > Question 4. The alignment algorithm in FuseCodec-ContextAlign appears heuristic. Have you considered more established sequence alignment methods from the speech community, such as using a forward-sum algorithm inspired by CTC or attention-based alignment, to derive a more principled matching between text and audio frames?
> >
> > **Response:**
> > We note that CTC-style forward-sum alignment, **while effective for ASR architectures**, is not directly applicable here. Incorporating such mechanisms **would require substantial architectural changes, including decoder-like components and loss constraints that fall outside the scope of a speech codec,** whose objective is discrete representation learning rather than sequence transcription. For this reason, we consider CTC-based alignment unnecessary for our setting.
> >
> > Regarding attention-based alignment: **FuseCodec-Fusion already integrate attention-based cross-modal interaction through our cross-attention module between semantic and contextual teacher signals.**

---

> ### Author Response · Authors · 2025-11-25
> **Methodological Fragmentation and Combining Methods (Part 2/6)**
>
> **Reviewer Comment:**
> > The paper's primary weakness is its methodological fragmentation. It presents three separate, mutually exclusive techniques (-Fusion, -Distill, -ContextAlign) rather than a single, principled framework. There is little to no discussion on why these three different approaches were chosen or how they might be combined.
>
> **Response:**
> We clarify that the three methods are not “fragmented,” but are **purposefully distinct**, each addressing **three different alignment challenges and limitations** in current speech tokenization.
>
> (i) Fusion: Fusion is our core **contribution for directly integrating acoustic + semantic + contextual information before quantization**. This has not been explored previously and directly resolves post-quantization alignment issues while improving reconstruction and representation performance.
>
> (ii) Distill: Baselines such as DM-Codec suffer from a fundamental problem: semantic and contextual embeddings come from different modalities and are **not temporally aligned** with RVQ tokens. **The global-pooled vectors provide an effective fix**: each RVQ token is guided by global sequence-level semantic/contextual vectors, avoiding the mismatched-token problem.
>
> (iii) ContextAlign: ContextAlign tackles a third issue: how to **leverage the full temporal structure of contextual embeddings while still maintaining alignment** with acoustic tokens. Our dynamic token-matching mechanism is simple, novel for this task, and enables temporally fine-grained alignment without relying on global pooling.
>
> **Each method solves a different failure mode** (latent fusion, global alignment, temporal alignment). They are therefore presented as standalone strategies, each strong on its own, and not as parts of a forced unified framework. In practice, one may choose the method depending on the desired properties of the tokenizer.
>
> **Reviewer Comment:**
> > Question 1. The three proposed methods are presented as separate variants. Did you attempt to combine them? For example, could one use latent fusion (-Fusion) and also apply temporally-aligned supervision (-ContextAlign)? If so, what were the results? If not, why is a unified approach not explored?
>
> **Response:**
> While it is technically possible to combine the methods, we intentionally chose not to. Fusion operates in the *pre-quantization space*, whereas Distill and ContextAlign operate in the *post-quantization space*. Stacking both levels of supervision **would inject overlapping semantic/contextual signals at multiple stages**, which is theoretically likely to **overfit the codec to supervised guidance** rather than letting the RVQ learn a robust discrete space.
>
> Moreover, our goal is not to force a single “everything combined” solution, but to provide **three principled alternatives/solutions that target different alignment issues**. Each variant already achieves strong standalone performance with no architectural complexity. For these reasons, we determined that exploring a unified hybrid approach was unnecessary and potentially counterproductive.

---

> ### Author Response · Authors · 2025-11-25
> **Variant-Specific Strengths (Part 3/6)**
>
> **Reviewer Comment:**
> > The results further highlight this issue: different variants excel on different tasks (e.g., -Fusion is best for reconstruction, -Distill for TTS intelligibility, -ContextAlign for TTS naturalness). This suggests an exploratory study of "what works" rather than a novel, unified contribution.
>
> **Response:**
> We respectfully disagree with the implication that a contribution must be “unified” in order to be novel. Each of our three methods is intentionally designed to **address a different failure mode in multimodal speech tokenizers**, and thus each stands as a meaningful contribution on its own.
>
> Fusion excelling in reconstruction, Distill in intelligibility, and ContextAlign in naturalness is not accidental. The observed **result pattern reflects the functional purpose of each technique:**
>
> **(i) Fusion** modifies the *latent pre-quantization space*, which naturally leads to improved reconstruction fidelity.
>
> **(ii) Distill** imposes *global semantic–contextual consistency*, which directly boosts TTS intelligibility.
>
> **(iii) ContextAlign** leverages *fine-grained temporal text guidance*, which is expected to improve TTS naturalness.
>
> **These variants were designed to be distinct, because the underlying challenges they address are distinct.** Attempting to force them into a single mechanism would risk blurring these roles rather than strengthening them. Our intention is to provide **three principled, targeted solutions**, each contributing to a key gap in multimodal speech tokenization.

---

> ### Author Response · Authors · 2025-11-25
> **Novelty and Global vs. Full-Sequence Guidance (Part 4/6)**
>
> **Reviewer Comment:**
> >    The core idea of distilling knowledge from pre-trained models (like HuBERT or BERT) into a codec is not new, as the paper itself acknowledges (citing works like SpeechTokenizer and DM-Codec). The novelty lies in the specific mechanisms, some of which are poorly justified: (1) The reliance on global vectors (mean-pooled semantics and a single [CLS] token for context) in both the -Fusion and -Distill variants is a major simplification. This approach discards vast amounts of temporal information inherent in speech and text. The paper fails to justify why this crude, global representation is preferable to using the full, time-aligned sequences from the teacher models. (2) The claims in Table 1 feel overstated. The distinctions between "Sim.", "Direct.", and "Align." are not clearly defined, making the table seem more like marketing than a rigorous technical comparison. For example, DM-Codec also performs alignment via similarity matching, so the checkmark system seems arbitrary.
>
> **Response:**
> We would like to clarify that although knowledge distillation has been utilized before in codecs, **our contributions address limitations that prior speech tokenization work has not resolved**.
>
> As stated in the manuscript, existing methods remain constrained in three key ways: (i) SpeechTokenizer lacks any contextual (text-derived) information, (ii) DM-Codec does not provide a principled/global mechanism for temporal alignment across modalities, and (iii) no prior work integrates semantic or contextual supervision directly into the latent pre-quantization space, which is a central innovation of our Fusion variant.
>
> Regarding the reviewer’s two points:
>
> **(1) Use of global semantic and contextual vectors:**
> The global vectors are not a simplification, but they are a **necessary correction** to a fundamental problem: **pre-trained semantic and contextual embeddings are not temporally aligned with each other or with RVQ tokens**. For example, the 10th semantic embedding, 10th contextual embedding, and 10th RVQ token do not correspond to the same underlying speech segment.
>
> Thus, directly using full sequences produces misaligned supervision. **Our global pooling resolves this by providing consistent, modality-aware guidance** to every RVQ token. In other words, each token learns with respect to the full semantic–contextual structure of the utterance, which is impossible with raw frame sequences.
>
> **(2) Distinctions in Table 1 (“Sim.”, “Direct.”, “Align.”):**
> We respectfully disagree that the distinctions are unclear. The table caption explicitly defines the categories: (i) **Sim.** – similarity-based supervision (ii) **Direct.** – direct latent integration, and (iii) **Align.** – modality alignment. DM-Codec does indeed use similarity matching (*and is correctly ticked under “Sim.”*), but it does not perform alignment across modalities and does not provide global semantic–contextual alignment or local matching. Moreover, no prior work performs direct fusion of external semantic/contextual signals into the encoder latent path before quantization. Thus, the table provides a **precisely defined technical comparison that highlights gaps in prior work and positions our contributions accordingly.**
>
> **Reviewer Comment:**
> > Question 2. Could you provide a stronger justification for using globally-pooled/CLS vectors for semantic and contextual guidance in -Fusion and -Distill? How does this compare, for instance, to using a standard cross-attention mechanism between the encoder's latent states and the full hidden state sequences from the teacher models?
>
>
> **Response:**
> Our use of globally pooled/CLS vectors is a **deliberate and necessary choice**, not a simplification.
> Direct cross-attention between the encoder’s latent states and the full teacher-model sequences is problematic because the two modalities have fundamentally different temporal resolutions.
>
> While cross-attention can operate on sequences of different lengths, it implicitly assumes that both sequences carry compatible temporal sequences. This assumption breaks in our setting: **contextual/text embeddings are always far shorter than semantic embeddings and encoder's latent embeddings** (e.g., ~20 text tokens vs. ~150 speech frames for a 3s utterance). Directly attending across such mismatched sequences computes attention for misaligned modality embeddings, and produces unstable/distorted attention maps.
>
> **Our globally pooled/CLS representations avoid this issue entirely. By distilling semantics and context into global vectors, we provide clean, modality-consistent guidance**. This results in more stable training and more predictable supervision compared to using an inherently mismatched sequence.

---

> ### Author Response · Authors · 2025-11-25
> **Magnitude of Improvements vs. Complexity (Part 5/6)**
>
> **Reviewer Comment:**
> > While the proposed methods achieve state-of-the-art results, the improvements over the strongest baselines are often marginal. For instance, in Table 2, FuseCodec-Fusion improves the WER over DM-Codec from 4.09 to 3.99 and ViSQOL from 3.20 to 3.47. These gains, while real, may not be significant enough to justify the substantial increase in system complexity. The training process requires running three large pre-trained models (ASR, LM, SSL) to generate guidance signals, which is computationally expensive.
>
> **Response:**
> We respectfully disagree that the improvements are marginal. Reconstruction and intelligibility metrics in speech tokenization are highly saturated, and even small absolute differences correspond to meaningful gains. For example, our ~10% relative WER improvement over DM-Codec is substantial given how tightly clustered prior work is. Similarly, the ViSQOL improvement over DM-Codec reflects a clear enhancement in perceptual quality.
>
> Regarding system complexity: our approach does not introduce a heavier training pipeline than **existing baselines. DM-Codec also relies on ASR/LM/SSL models to produce guidance signals. Our contribution operates within this established framework by addressing its limitations (misalignment, lack of direct integration),** rather than adding new large components.
>
> Importantly, there is **no notable increase in runtime complexity:** all semantic/contextual guidance signals are pre-computed once and cached. This takes roughly 1 second per utterance (about 2.5 hours for LibriSpeech-100h). Our method does not involve/require running ASR/LM/SSL models during codec training. The training-time overhead is minimal, **limited to lightweight operations such as tensor addition (for fusion) and a loss computation (for supervision)**, which do not significantly impact training time or scalability.
>
> **Reviewer Comment:**
> > Question 3. The "FuseCodec (Baseline)" in Table 2 performs worse than several existing codecs. Could you elaborate on this baseline's architecture? Is it a reproduction of a standard model like EnCodec? A stronger, more competitive baseline would better isolate the true contribution of your proposed fusion and supervision techniques.
>
> **Response:**
> As noted in the manuscript (lines 255–258), the FuseCodec (Baseline) uses the **same standard architecture as established models such as EnCodec, SpeechTokenizer, and DM-Codec.** *Full architectural details are provided in Appendix E.1,* and the implementation is an almost one-to-one reproduction of the conventional codec design used across prior work.
>
> This was intentional: our goal is to isolate the true contribution of our fusion, supervision, and alignment strategies on top of a widely accepted backbone. Moreover, **Table 2 already compares our variants against 10 strong, competitive codec baselines**, ensuring that improvements are measured against robust and strong baselines. Please refer to section 3.2.1.

---

### Official Review · Reviewer_Bd29 · 2025-10-31

**Soundness:** 2
**Presentation:** 2
**Contribution:** 2
**Rating:** 2
**Confidence:** 5

**Summary:**

This paper proposes FuseCodec, a neural audio codec that aims to integrate acoustic, semantic, and contextual information through three complementary strategies: latent representation fusion, global semantic-contextual supervision, and temporally aligned contextual supervision. The idea is to enrich discrete codec representations using multimodal cues from pretrained self-supervised speech and language models. The work is positioned as an extension over recent multimodal codecs such as DM-Codec and SpeechTokenizer, aiming to provide tighter cross-modal alignment and improved downstream performance.

While the motivation of unifying contextual and semantic representations is sound, the technical contribution appears incremental and heavily engineered. The simplest variant performs best—latent fusion of global vectors. The added complexity of the global and temporally aligned supervision variants does not clearly translate into consistent or interpretable gains. In particular, I would have expected ContextAlign to improve content preservation but yields higher WER and only better UTMOS, which might indicate that perceptual quality improves independently of alignment accuracy. The experimental evidence does not fully support the stated motivation for complex alignment schemes. Moreover, the paper lacks a minimal baseline comparison using direct latent fusion of global semantic-context vectors without cross-attention or other modules, which would clarify the necessity of the proposed mechanisms.

**Strengths:**

1. Attempt to unify acoustic, semantic, and contextual modeling within a codec framework is interesting.

2. Evaluation across reconstruction, representation quality, and downstream TTS tasks.

3. Comparison with strong baselines (EnCodec, SpeechTokenizer, DM-Codec, etc.).

4. Demonstrates consistent improvements for the fusion variant on several benchmarks and bitrate-efficient design (4 kbps).

**Weaknesses:**

1. Most improvements come from latent fusion, while the additional supervision/align variants add complexity without proportional gains.

2. The claim of “temporal dynamics” from global pooled vectors is conceptually unclear; global mean-pooled embeddings carry no temporal information.

3. Cross-attention between already global vectors seems unnecessary and not well justified.

4. ContextAlign underperforms in WER/WIL despite being motivated as improving alignment.

5. Missing comparison against a simpler baseline that fuses global semantic and contextual embeddings directly without extra attention or alignment or supervision layers.

6. Limited discussion of statistical significance of the results, no confidence intervals for any of the metrics.

**Questions:**

1. How does cross-attention between globally pooled vectors meaningfully contribute beyond simple concatenation or addition?

2. Why is the “temporal dynamics” claim justified when both semantic and contextual signals are globally aggregated?

3. Can you report a baseline with direct latent fusion of global semantic/contextual vectors without cross-attention or supervision losses?

4. For the ContextAlign variant, what explains the significantly higher WER yet better UTMOS? There are no failure case audio samples in the supplementary materials.

---

> ### Author Response · Authors · 2025-11-25
> **Summary of Responses to Reviewer Bd29’s comments (Part 1/7)**
>
> We thank the reviewer for recognizing our attempt to unify acoustic, semantic, and contextual modeling within a codec framework, the comprehensive evaluation across reconstruction, representation, and downstream TTS, the strong baseline comparisons, and that the fusion variant demonstrates consistent improvements within a bitrate-efficient design.
>
> **Summary of Author Responses:**
>
> - We clarify the question about added complexity by noting that Distill and ContextAlign introduce no architectural changes or parameters (only lightweight cosine-similarity losses), while yielding proportional improvements over baselines.
>
> - We address the question about “temporal dynamics” by explaining that global pooled vectors do not contain temporal dynamics; instead, they shape temporal dynamics / token learning through supervision.
>
> - We explain why cross-attention between global vectors is required by noting that simple addition underperforms, and our ablations show that cross-attention meaningfully improves metrics.
>
> - We clarify the ContextAlign WER/UTMOS pattern by showing that its objective prioritizes local acoustic consistency (benefiting UTMOS) rather than global semantic structure (affecting WER), and that it still surpasses many strong baselines in both WER and WIL.
>
> - We respond to the request for a simpler fusion baseline by noting we ablated this: FuseCodec-Fusion (Attn-Proj: None). This direct fusion baseline is outperformed by variants using attention.
>
> - We justify not reporting confidence intervals as neural codecs are deterministic pipelines (codebook matching) with no sampling variance across runs, and we match the evaluation practice standard in speech tokenization literature.
>
> Below we clarify regarding complexity, global vectors, cross-attention, alignment performance, baseline comparisons, and statistical reporting.

---

> > ### Author Response · Authors · 2025-11-25
> > **Complexity and Gains (Part 2/7)**
> >
> > **Reviewer Comment:**
> > > Most improvements come from latent fusion, while the additional supervision/align variants add complexity without proportional gains.
> >
> > **Response:**
> > We clarify that the proposed supervision and alignment variants are *not complex*. Our method introduces **no architectural changes and no additional parameters** compared to SpeechTokenizer or the DM-Codec baseline.
> > Instead, our methodology is implemented as a simple cosine-similarity-based loss. The Distill variant uses mean-pooled semantic and the [CLS] token contextual embeddings for supervision. The ContextAlign variant applies a lightweight windowed cosine-matching for temporal alignment. These are **standard, inexpensive operations that improve the supervision/alignment procedure without adding meaningful complexity.**
> >
> > Moreover, the **resulting gains are significant and fully proportional** to any added complexity. In Table 2 (Sec 3.2.1), FuseCodec-Distill alone surpasses strong baselines. For example PESQ 3.06 vs. 2.77 (DM-Codec), 2.68 (BigCodec), 2.60 (SpeechTokenizer) and ViSQOL 3.43 vs. 3.36 (DAC), 3.20 (DM-Codec), while also achieving top-tier UTMOS 3.65.
> >
> > Similarly, **FuseCodec-ContextAlign outperforms major baselines** with lower WER (6.70) and WIL (4.15), and higher STOI (0.94), ViSQOL (3.18), and PESQ (2.85) against Baselines: BigCodec, Mimi, SpeechTokenizer, StableCodec, WavTokenizer, X-Codec2. These results highlight the **improvements achieved even without Fusion** strategy.

---

> > ### Author Response · Authors · 2025-11-25
> > **Necessity of Cross-Attention (Part 4/7)**
> >
> > **Reviewer Comment:**
> > > Cross-attention between already global vectors seems unnecessary and not well justified.
> > >> Question 1. How does cross-attention between globally pooled vectors meaningfully contribute beyond simple concatenation or addition?
> >
> > **Response:**
> > We emphasize that the use of cross-attention between our two global vectors is **both empirically justified and methodologically necessary**. *Our ablation study in Section D.1 (“Ablation: Attention–Projection Configuration in Representation Fusion”)* explicitly shows that applying **cross-attention to the global vectors before projection significantly improves performance compared to simple addition** (None vs. Cross-Before configuration).
> >
> > Conceptually, cross-attention **allows us to capture interdependencies between the two different modality vectors** → semantics from speech and context from text.  Rather than simply adding signals, our approach lets these vectors query and update each other first, resulting in more aligned and mutually informed vectors.
> >
> > This idea is supported by prior works in the speech domain, although in automatic speech recognition tasks. For instance, [1] **merge a text-level global context into the decoder via cross-attention** to substantially improve ASR performance.
> >
> > Reference:
> >
> > [1] Qian, Yukun & Zhuang, Xuyi & Wang, Mingjiang. (2023). Automatic Speech Recognition Transformer with Global Contextual Information Decoder. 4474-4478. 10.21437/Interspeech.2023-1450.

---

> > ### Author Response · Authors · 2025-11-25
> > **Statistical Significance (Part 7/7)**
> >
> > **Reviewer Comment:**
> > >    Limited discussion of statistical significance of the results, no confidence intervals for any of the metrics.
> >
> > **Response:**
> > We respectfully note that **confidence intervals are not typically reported for neural speech codecs**, because, unlike stochastic TTS or generative language models, **neural codecs produce deterministic outputs from codebooks**. Given a fixed input waveform and fixed codebook, the encoder-quantizer-decoder pipeline yields identical reconstruction on every run.
> >
> > For example, once an RVQ codebook is learned, the quantizer performs:
> >  $
> >  z_t = \arg\min_k \lVert x_t - c_k \rVert
> >  $
> >
> > This produces **the same code index every time** for a given frame. The resulting reconstruction is therefore stable and statistically invariant, making interval-based reporting unnecessary.
> >
> > Moreover, this evaluation protocol is standard across baseline codec papers (EnCodec [1], SpeechTokenizer [2], XCodec2 [3], BigCodec [4]): objective metrics are single deterministic measurements. For these reasons, we believe the reported metrics already fully characterize performance without requiring confidence intervals.
> >
> > Reference:
> >
> > [1] https://arxiv.org/abs/2210.13438
> >
> > [2] https://arxiv.org/abs/2308.16692
> >
> > [3] https://arxiv.org/abs/2502.04128
> >
> > [4] https://arxiv.org/abs/2409.05377

---

> ### Author Response · Authors · 2025-11-25
> **Temporal Dynamics Explanation (Part 3/7)**
>
> **Reviewer Comment:**
> > The claim of “temporal dynamics” from global pooled vectors is conceptually unclear; global mean-pooled embeddings carry no temporal information.
> >> Question 2. Why is the “temporal dynamics” claim justified when both semantic and contextual signals are globally aggregated?
>
> **Response:**
> We clarify that the “temporal dynamics” statement does not claim that the global pooled vectors (mean-pooled semantic vector or the [CLS] contextual vector) inherently contain temporal information. Rather, **it refers to how these global embeddings guide the learning of each RVQ token across time** (lines 203–204).
>
> In our supervision, every RVQ token learns its representation while being informed by the global sequence-level semantic and contextual signals. For example, a vowel sound in one token can adjust its representation based on the global sentence-level semantic and contextual cues. In this sense, the **global signals influence how “temporal dynamics” emerge in the learned discrete codes.**
>
> This mechanism is consistent with prior work on how global objectives shape frame-level temporal representations. For example, HuBERT [1] uses non-temporal k-means cluster assignments as supervision for every masked frame, stating: *“Concretely, a BERT model consumes masked continuous speech features to predict pre-determined cluster assignments.”* This **demonstrates that global/non-temporal targets drive temporally coherent frame-level learning through supervision**.
>
> Reference:
>
> [1] Wei-Ning Hsu, Benjamin Bolte, Yao-Hung Hubert Tsai, Kushal Lakhotia, Ruslan Salakhutdinov, and Abdelrahman Mohamed. Hubert: Self-supervised speech representation learning by masked prediction of hidden units. IEEE/ACM transactions on audio, speech, and language processing, 29:3451–3460, 2021.

---

> ### Author Response · Authors · 2025-11-25
> **ContextAlign WER vs. UTMOS (Part 5/7)**
>
> **Reviewer Comment:**
> >  ContextAlign underperforms in WER/WIL despite being motivated as improving alignment.
> >>  Question 4. For the ContextAlign variant, what explains the significantly higher WER yet better UTMOS? There are no failure case audio samples in the supplementary materials.
>
> **Response:**
> We clarify that the behavior of ContextAlign is expected given its objective. ContextAlign provides **fine-grained, timestep-level contextual supervision**, focusing on precise local alignment between RVQ tokens and text embeddings. **Because UTMOS is a perceptual quality metric that evaluates short-time acoustics, any improvement in local acoustic consistency directly increases its score.**
>
> However, because ContextAlign does not inject global semantic signals (unlike Fusion or Distill), it offers less guidance on sentence-level linguistic structure. This naturally leads to higher WER/WIL compared to our variants that explicitly supervise global semantics and contexts.
>
> In Table 2 (Section 3.2.1), **FuseCodec-ContextAlign outperforms major baselines** with lower WER (6.70) and WIL (4.15), and higher STOI (0.94), ViSQOL (3.18), and PESQ (2.85) against Baselines: BigCodec, Mimi, SpeechTokenizer, StableCodec, WavTokenizer, X-Codec2. These results highlight the **aligned contextual supervision is effective even without global semantic or fusion.**

---

> ### Author Response · Authors · 2025-11-25
> **Simple Fusion Baseline (Part 6/7)**
>
> **Reviewer Comment:**
> > Missing comparison against a simpler baseline that fuses global semantic and contextual embeddings directly without extra attention or alignment or supervision layers.
> >> Question 3. Can you report a baseline with direct latent fusion of global semantic/contextual vectors without cross-attention or supervision losses?
>
> **Response:**
> We already include this baseline in our ablation study in *Section D.1 (“Ablation: Attention–Projection Configuration in Representation Fusion”)*. The variant **FuseCodec-Fusion (Attn-Proj: None) corresponds exactly to the reviewer’s request**: it fuses the global semantic and contextual vectors directly into the latent space **without any cross-attention, alignment, or supervision losses.**
>
> **FuseCodec variants with attention consistently surpass the simple baseline** (4.10 WER, 6.60 WIL, 3.26 ViSQOL, 2.92 PESQ). In particular, FuseCodec (Attn-Proj: Cross-Before), which adds a cross-attention layer before projection, shows clear gains across all major metrics, with lower WER (3.99 vs. 4.10), lower WIL (6.45 vs. 6.60), higher ViSQOL (3.47 vs. 3.26), and higher PESQ (3.13 vs. 2.92). Please refer to the Table 6 (ablation study in Section D.1.).
>
> This direct comparison demonstrates that **adding cross-attention meaningfully strengthens both content preservation and perceptual quality**, confirming the necessity of the proposed design beyond simple fusion.

---

> ### Comment · Reviewer_Bd29 · 2025-11-27
> **Response to Author Comments**
>
> Thanks for the detailed response and clarifications.
> First of all I would like to state that I had misunderstood the fusion method a bit, and I was under the impression that the fusion was always applied and on top of that the other two variants were built. This confusion mainly came from the fact that in the diagram, because the RVQ block is colored the same and there are no separate notations for the quantized vectors with and without the fusion.
>
> With this misunderstanding cleared, I have a better realization of the impact of the other two methods.
>
> Thank you for the clarification regarding the cross-attention fusion module. My concern is not only empirical but structural — when the semantic and contextual vectors are broadcast across the time dimension, the cross-attention operation collapses to a global nonlinear transform rather than performing token-level or alignment-based fusion.
>
> Let \\(\hat S \in \mathbb{R}^D\\) and \\(\hat C \in \mathbb{R}^D\\) be the global semantic and contextual vectors that are then broadcast:
>
> \\[
> \tilde S_t = \hat S,\qquad \tilde C_t = \hat C,\qquad t=1,\dots,T'.
> \\]
>
> For the Cross-Before block:
>
> \\[
> S' = \\text{CrossAttn}(\tilde S,\tilde C,\tilde C)W_S,
> \\]
>
> the internal attention linear projections are:
>
> \\[
> Q_t = \hat S W_Q,\qquad K_i = \hat C W_K,\qquad V_i = \hat C W_V.
> \\]
>
> Since all queries and keys are identical across all timesteps:
>
> \\[
> Q_t K_i^\\top = \\text{const} \Rightarrow \alpha_{t,i}=\frac{1}{T'}.
> \\]
>
> Therefore the attention output becomes:
>
> \\[
> S_t = \sum_{i} \alpha_{t,i} V_i W_S
> \\]  (for some reason the S'_t does not render correctly, in the above equation read S_t as S'_t)
>
> \\[
> S'_t = f(\hat C) W_S,
> \\]
>
> independent of \\(\hat S\\) and independent of time. Symmetrically:
>
> \\[
> C'_t = g(\hat S) W_C.
> \\]
>
> Thus the two cross-attention blocks reduce to the form:
>
> \\[
> S' = f(\hat C)W_S,\qquad C' = g(\hat S)W_C,
> \\]
>
> which are then fused with the latent variables:
>
> \\[
> Z'_t = Z_t + f(\hat C)W_S + g(\hat S)W_C.
> \\]
>
> This implies that cross-attention does not perform token-wise or fine-grained multimodal alignment. It acts as a global nonlinear feature fusion layer applied uniformly across time. The performance improvement over the None variant appears to be attributable to additional projection capacity and a richer transformation \\(f\\) and \\(g\\), rather than to attention-specific behavior. In other words, cross-attention is not uniquely necessary in this formulation, as other global fusion functions (FiLM, gated MLPs, bilinear mixing, etc.) could represent the same family of transformations without broadcasting or attention.
>
> While the ablation confirms that multimodal fusion helps, the formulation does not yet fully justify the necessity of cross-attention itself (as opposed to simply using a stronger global fusion mechanism).
>
> Additionally, since all three methods also bring significant improvement over the other baselines, and the fusion version performs the best, I think applying the other two variants on top of fusion could potentially lead to further improvements.
>
> I will increase my score from 2 → 4 since one of my major misunderstandings is cleared; however my concerns regarding the necessity of cross-attention for global vectors is still not resolved.

---

> > ### Author Response · Authors · 2025-11-29
> > **Necessity of Cross-Attention and Non-Stacking Variants**
> >
> > **Reviewer Comment:**
> > > ... While the ablation confirms that multimodal fusion helps, the formulation does not yet fully justify the necessity of cross-attention itself (as opposed to simply using a stronger global fusion mechanism).
> >
> > **Response:**
> > Thank you for the follow-up, for revisiting the methodology, and for the detailed mathematical analysis.
> > We agree that broadcasting global semantic/contextual vectors results in constant attention weights. However, **cross-attention remains the appropriate choice for Fusion because:**
> >
> > **(1) Structured multimodal transformation is required before injecting global semantics/context into encoder latents.** Our goal in Fusion is to transform and inject *global* semantic+contextual info (global vectors) *uniformly* into the encoder latent space. These three modalities (Tokenizer latents, HuBERT semantics, BERT context) come from **inherently different representation spaces, and making them compatible requires a structured multi-projection and transformation mechanism**, which the Cross-attention provides:
> > $
> > S' = f(\hat C)\,W_S, \qquad
> > C' = g(\hat S)\,W_C
> > $
> >
> > **Alternatively, a single projection or gating function would treat all modalities identically and cannot learn two distinct cross-modal mappings simultaneously without explicit duplication**, since a single operator would take the form:
> > $
> > h([\hat S \,\|\, \hat C])
> > $
> >
> > **(2) Cross-attention enables bidirectional cross-modal conditioning.** Fusion performs both *semantic→contextual* and *contextual→semantic* conditioning before integration with encoder latents. **The Q/K/V formulation naturally provides two distinct learned pathways for these directions.**
> >
> > $
> > \text{Attn}(Q_S, K_C, V_C)
> > \qquad\text{and}\qquad
> > \text{Attn}(Q_C, K_S, V_S)
> > $
> >
> > Achieving the same behavior with **FiLM, gated MLPs, or bilinear mixing would require manually duplicating and coordinating multiple parallel pathways**, reducing modularity and limiting expressiveness.
> >
> > **(3) Multi-head attention offers modality factorization and richer global transformations.** Because our three modality embeddings have different representational properties, **multi-head attention** (8 heads in our implementation) **decomposes the fusion into multiple learned subspaces:**
> > $
> > S' = \sum_{h=1}^{H} f_h(\hat C)\, W_S^{(h)}, \qquad
> > C' = \sum_{h=1}^{H} g_h(\hat S)\, W_C^{(h)}
> > $
> >
> > Here, each head $h$ specializes in a distinct interaction pattern. Operators such as **FiLM, gated MLPs, or bilinear mixing apply one uniform global transformation and cannot factorize modalities or learn multiple interaction patterns simultaneously.**
> >
> > For these reasons, Cross-Attention is an appropriate and effective mechanism for fusing global semantic/contextual information. While other global fusion operators (FiLM, gated MLPs, bilinear mixing, etc.) could in principle represent similar global transformations, this does not diminish **the empirical advantage we observe from attention in our setting; its consistently superior performance justifies its inclusion. Exploring alternative operators is reasonable future work, but not required for the validity of our approach.**
> >
> > **Reviewer Comment:**
> > > Additionally, since all three methods also bring significant improvement over the other baselines, and the fusion version performs the best, I think applying the other two variants on top of fusion could potentially lead to further improvements.
> >
> > **Response:**
> > Although stacking the variants might seem beneficial, **the three methods/variants are intentionally distinct, each addressing a different alignment constraint in speech tokenization.** *Fusion operates pre-quantization, modifying encoder latents*; *Distill and ContextAlign operate post-quantization, supervising or aligning RVQ tokens*.
> >
> > Combining them would inject semantic/contextual signals at multiple stages, creating overlapping supervision that risks overconstraining or overfitting the discrete space rather than letting RVQ learn robust tokens.
> > For these reasons, we determined that a hybrid stacked version is neither necessary nor theoretically advantageous. Please see the detailed response here: Methodological Fragmentation and Combining Methods (Part 2/6) -- https://openreview.net/forum?id=RILri9w9IW&noteId=kDoWkwax2U

---

### Official Review · Reviewer_mcfd · 2025-11-01

**Soundness:** 3
**Presentation:** 3
**Contribution:** 3
**Rating:** 4
**Confidence:** 3

**Summary:**

The paper proposes FuseCodec, a speech tokenization framework that enriches neural codec tokens with semantic (SSL speech) and contextual (LM) information using three complementary techniques: (i) Latent Representation Fusion (injects semantic/contextual guidance into encoder latents before RVQ), (ii) Global Semantic-Contextual Supervision (distills pooled semantic/context vectors into first-stage RVQ tokens), and (iii) Temporally Aligned Contextual Supervision (a local window matching algorithm to align RVQ tokens with LM tokens for token-level losses). Using frozen wav2vec2 (ASR), BERT (LM), and HuBERT (SSL), the method reports SOTA on LibriSpeech reconstruction (intelligibility, quality, speaker similarity) and strong results on Codec-SUPERB at 4 kbps, and demonstrates applicability to zero-shot TTS.

**Strengths:**

- This paper organizes prior semantic/contextual guidance ideas into a single codec framework with three variants that are easy to ablate and combine.

- The windowed token-matching for contextual supervision is simple, inexpensive, and addresses speech/text rate mismatch better than global pooling.

- The authors report ASR WER/CER, STOI, perceptual (NISQA/UTMOS), speaker similarity, and downstream TTS. These results show evidence that the learned tokens carry useful information beyond acoustics.

**Weaknesses:**

- My main concern is that the paper is purely engineering and no theoretical justification. The window update heuristic (dynamic shift) is intuitive but somewhat ad-hoc; no analysis of stability/complexity vs sequence length or error propagation when early matches are wrong.

- The models are trained primarily on LibriSpeech/LibriTTS (English, read speech). It is unclear whether gains persist on conversational, noisy, or non-English corpora; contextual alignment may degrade with ASR errors and long-form prosody.

- Improvements may hinge on the quality of the ASR/LM/SSL guides (w2v2/BERT/HuBERT). The paper does not quantify sensitivity to these choices or to ASR transcription errors used to obtain LM tokens.

- While components are described, it is not fully clear (a) how much each of Fusion vs Global-Distill vs ContextAlign contributes under identical training budgets, (b) whether Global-Distill + ContextAlign are complementary or redundant, and (c) how performance scales with RVQ depth/codebook size and 50 Hz frame rate.

**Questions:**

- How does ContextAlign perform when transcripts are noisy (e.g., phone ASR with 15–20% WER)? Any results with weaker/stronger ASR or end-to-end speech-LM context (without text)?

- What happens if we replace BERT with a recent LLM or a Conformer LM, or HuBERT with wavLM/Wav2Vec2-Large?

---

> ### Author Response · Authors · 2025-11-25
> **Summary of Responses to Reviewer mcfd’s Comments (Part 1/7)**
>
> We thank the reviewer for the thoughtful analysis. We appreciate the recognition that our work brings together prior semantic and contextual guidance ideas into a unified codec framework, and that the three variants: Fusion, Global-Distill, and ContextAlign, are organized clearly and are easy to ablate. We are glad the reviewer highlighted that the windowed token-matching strategy is simple, inexpensive, and handles the speech/text rate mismatch effectively. We also appreciate the acknowledgement that our learned tokens carry information beyond acoustics, as evidenced by our evaluations across intelligibility, perceptual quality, speaker similarity, and downstream TTS, and that our overall goal of enriching codec tokens with semantic and contextual cues is interesting and well motivated.
>
> **Summary of Author Response:**
>
> - We clarify the theoretical motivation by explaining how each variant addresses a specific weakness in current codecs: lack of pre-quantization fusion, modality mismatch, and need for local temporal conditioning.
>
> - We clarify the window heuristic by explaining that it follows the natural many-to-one mapping between speech frames and text tokens and remains stable through re-evaluation within a bounded window.
>
> - We address generalization concerns by pointing to strong performance across Codec-SUPERB’s conversational, noisy, and multilingual datasets, as well as unseen multilingual evaluation.
>
> - We provide additional results assessing SSL/LM choices by showing stable performance when replacing HuBERT/BERT with wav2vec2-large, wavLM, E5, or Electra.
>
> - We provide additional results evaluating transcript-noise sensitivity using a weaker/smaller ASR model, showing only modest degradation even under heavily degraded transcripts.
>
> - We discuss the contributions and scalability by showing identical training budgets, roles of Distill and ContextAlign, and improved performance with larger RVQ/frame-rate configurations.
>
> Below, we address the reviewer’s concerns in detail.

---

> > ### Author Response · Authors · 2025-11-25
> > **Theoretical Motivation (Part 2/7)**
> >
> > **Reviewer Comment:**
> > > My main concern is that the paper is purely engineering and no theoretical justification.
> >
> > **Response:**
> > We clarify that each of the three proposed variants is grounded in theoretical needs arising from limitations in current speech tokenization systems:
> >
> > **(i) Fusion: addressing missing pre-quantization multimodal integration.**
> >  Prior work injects semantic information after quantization, leaving the alignment bottleneck unresolved. **Our Fusion approach is theoretically required** because by the principle of the quantization bottleneck, only pre-quantization signals can shape the discrete codebook. Thus, integrating semantic and contextual cues before RVQ is not an engineering choice but a necessity for multimodal code learning.
> >
> > **(ii) Distill: addressing modality-resolution mismatch.**
> >  Semantic (SSL) and contextual (LM) embeddings have incompatible temporal lengths. **Global pooling is theoretically required to avoid mismatched supervision**: it creates a sequence-level global target that every RVQ token can reliably learn from, bypassing complex/unsuitable one-to-one alignment assumptions.
> >
> > **(iii) ContextAlign: addressing local temporal conditioning.**
> > Global pooling removes local detail. **ContextAlign is theoretically grounded** in the monotonic many-to-one mapping between speech frames and text tokens: multiple acoustic frames correspond to a single linguistic unit. Our lightweight windowed matching satisfies monotonicity and avoids forcing frame-level alignment while still providing local contextual guidance.

---

> > ### Author Response · Authors · 2025-11-25
> > **Window Update Heuristic (Part 3/7)**
> >
> > **Reviewer Comment:**
> > > The window update heuristic (dynamic shift) is intuitive but somewhat ad-hoc; no analysis of stability/complexity vs sequence length or error propagation when early matches are wrong.
> >
> > **Response:**
> > We clarify that the dynamic-shift window heuristic is not ad-hoc, but a **theoretically grounded solution** to the inherent mismatch between text-token sequences and RVQ-frame sequences. In speech, *one text token (from transcriptions) typically corresponds to multiple acoustic frames*; thus, a local window is necessary for meaningful alignment.
> >
> > Our method simply identifies the highest-similarity region for each contextual token and **shifts the window forward accordingly**. Because the matching is driven by cosine similarity, the *risk of early wrong matches propagating is low*, and **each step re-evaluates similarity within a bounded window again**, making the process stable.
> >
> > For completeness, we include our **Fixed vs. Dynamic Window Configuration** ablation (Appendix D.3, Table 8) here. **Dynamic variants adapt the alignment window based on semantic/contextual similarity, whereas fixed variants use a rigid uniform window.** Dynamic windowing consistently yields **better intelligibility and clarity**, confirming that our alignment strategy is both stable and beneficial.
> >
> > ## Ablation: Dynamic vs. Fixed Windowing Under Different Guidance Modes
> > Dynamic = per-token adaptive window; Fixed = uniform window; Semantic-Contextual = combined guidance.
> >
> > | Model Variant              | Window      | Guidance                |   WER↓       |   WIL↓       |   STOI↑      |  ViSQOL↑     |   PESQ↑      |  UTMOS↑      | Similarity↑     |
> > |------------------------    |---------    |---------------------    |:--------:    |:--------:    |:--------:    |:--------:    |:--------:    |:--------:    |:-----------:    |
> > | FuseCodec-ContextAlign     | Fixed       | Contextual              |   4.26       |   6.88       |   0.92       | **3.19**     |   2.71       |   3.58       |    0.994        |
> > | FuseCodec-ContextAlign     | Dynamic     | Contextual              | **4.15**     | **6.70**     | **0.93**     |  _3.18_      | **2.85**     |   3.65       |    0.995        |
> > | FuseCodec-ContextAlign     | Fixed       | Semantic-Contextual     |   4.30       |   6.88       |   0.92       |   3.10       |   2.62       |  _3.74_      |    0.995        |
> > | FuseCodec-ContextAlign     | Dynamic     | Semantic-Contextual     |  _4.21_      |  _6.78_      |  _0.93_      |   3.12       |  _2.72_      | **3.75**     |    0.995        |
> >
> > **Summary of Results:**
> >
> > - Dynamic windowing is structurally required to match text token durations to RVQ frames. Also, windowing is stable because similarity is recomputed at every step, wrong early matches do not cascade.
> >
> > - Results demonstrates that the dynamic window mechanism is robust, stable, and beneficial, not prone to error propagation. In all cases, **Dynamic > Fixed**, improving WER/WIL and perceptual metrics (PESQ, UTMOS).
> >
> >
> > Furthermore, the **computational complexity remains minimal**: the method uses lightweight similarity computations over a small sliding window, adding no meaningful cost relative to standard distillation procedures.

---

> > ### Author Response · Authors · 2025-11-25
> > **Training Data and Generalization (Part 4/7)**
> >
> > **Reviewer Comment:**
> > > The models are trained primarily on LibriSpeech/LibriTTS (English, read speech). It is unclear whether gains persist on conversational, noisy, or non-English corpora; contextual alignment may degrade with ASR errors and long-form prosody.
> >
> > **Response:**
> > We agree that LibriSpeech/LibriTTS is English read speech; however, this is the standard and exclusive training setup used in our baseline codec works (EnCodec [1], SpeechTokenizer [2], DM-Codec [3], X-Codec2 [4], BigCodec [5]). Following the similar evaluation protocol, we intentionally use the same dataset to ensure a **fair and controlled comparison**, isolating the improvements that come purely from our fusion/supervision/alignment strategies.
> >
> > Despite this, **FuseCodec already demonstrates strong generalization**. **Codec-SUPERB spans 20 diverse datasets, including conversational (VoxCeleb1, IEMOCAP), noisy/overlapped (Libri2Mix, LibriCount, QUESST), and non-English speech (VoxLingua107, QUESST).** FuseCodec achieves strong results across these settings, even without multilingual or noisy training data (See Table 3: Representation quality results on the manuscript.)
> >
> > Additionally, our **appendix includes unseen multilingual speech evaluation** (See Appendix C.1) tests FuseCodec on speech reconstruction across seven unseen languages. The results show FuseCodec-Fusion achieves the strongest content and perceptual scores, with Distill maintaining second-best performance. Thus, although trained on standard English read speech, FuseCodec **generalizes broadly**, and the reviewer’s concern is empirically addressed.
> >
> > References:
> >
> > [1] https://arxiv.org/abs/2210.13438
> >
> > [2] https://arxiv.org/abs/2308.16692
> >
> > [3] https://arxiv.org/abs/2410.15017
> >
> > [4] https://arxiv.org/abs/2502.04128
> >
> > [5] https://arxiv.org/abs/2409.05377

---

> > ### Author Response · Authors · 2025-11-25
> > **Sensitivity to Teacher Models (Part 5/7)**
> >
> > **Reviewer Comment:**
> > > Improvements may hinge on the quality of the ASR/LM/SSL guides (w2v2/BERT/HuBERT). The paper does not quantify sensitivity to these choices or to ASR transcription errors used to obtain LM tokens.
> >
> > **Response:**
> > We note that pretrained models such as wav2vec 2.0, HuBERT, and BERT already provide **high-quality, robust representations**, used similarly in prior distillation-based codecs (SpeechTokenizer, DM-Codec). Our methods follow the same established practice, but **we solve the core limitations (temporal misalignment and lack of direct integration), rather than introducing new dependencies.**
> >
> > Given that the fused/distilled representations consistently improve reconstruction, representation quality, and downstream performance by **using same teacher models as earlier works**, we conclude that the gains stem from our alignment and integration strategies, not from sensitivity to the teacher models themselves. Please refer to the comparisons with SpeechTokenizer and DM-Codec in Tables 2-4 (Sections 3.2.1-3.2.3), which show that improvements persist under identical teacher model settings compared to these baselines.

---

> > ### Author Response · Authors · 2025-11-25
> > **Robustness to Noisy Transcripts (Part 6/7)**
> >
> > **Reviewer Comment:**
> > > Question 1. How does ContextAlign perform when transcripts are noisy (e.g., phone ASR with 15–20% WER)? Any results with weaker/stronger ASR or end-to-end speech-LM context (without text)?
> >
> > **Response:**
> > **FuseCodec-ContextAlign is unaffected by noisy transcripts at inference time**, because **no ASR is required during inference/testing**. Transcripts are only used *once* during training to obtain contextual embeddings from a frozen LM. The ASR component is fully replaceable, and datasets with transcripts (e.g., Emilia) would require no ASR at all.
> >
> > If the training transcripts themselves are noisy, performance can naturally decline. **However, this should not be an issue in actual training**, as high-quality transcripts are easily available for other speech datasets (e.g., Emilia). For completeness, we trained a **FuseCodec-ContextAlign-BadASR** variant using wav2vec2-base-100h transcripts (≈50% worse WER compared to wav2vec2-base-960h). The degradation is modest considering the noise level, and even with heavily degraded transcripts the model **remains stable and functional**:
> > | Model                             | ASR Model              | WER ↓     | WIL ↓     | STOI ↑     | ViSQOL ↑     | PESQ ↑     | UTMOS ↑      | Similarity ↑     |
> > |-------------------------------    |--------------------    |-------    |-------    |--------    |----------    |--------    |----------    |--------------    |
> > | FuseCodec-ContextAlign            | wav2vec2-base-960h     | 4.15      | 6.70      | 0.93       | 3.18         | 2.85       | 3.65    | 0.995            |
> > | FuseCodec-ContextAlign-BadASR     | wav2vec2-base-100h     | 6.02      | 9.08      | 0.90       | 2.64         | 2.19       | 2.81         | 0.982            |

---

> > ### Author Response · Authors · 2025-12-04
> > **Component Contributions and Scalability (Part 8)**
> >
> > **Reviewer Comment:**
> > > While components are described, it is not fully clear (a) how much each of Fusion vs Global-Distill vs ContextAlign contributes under identical training budgets, (b) whether Global-Distill + ContextAlign are complementary or redundant, and (c) how performance scales with RVQ depth/codebook size and 50 Hz frame rate.
> >
> > **Response:**
> > **(a) Contribution under identical training budgets.**
> > All FuseCode variants (Fusion, Global-Distill, and ContextAlign) are trained under identical training budgets (same data, same number of steps, same model size). Their relative improvements, therefore, reflect the contribution of the method itself.
> >
> > **(b) Complementary or redundant.**
> > Distill and ContextAlign are **purposefully distinct** mechanisms designed to address different forms of alignment mismatch:
> >
> > i. Distill aligns RVQ tokens with **global semantic/contextual embeddings**. This resolves modality mismatch and stabilizes supervision by ensuring every RVQ token is guided by a consistent sequence-level target.
> >
> > ii. ContextAlign instead uses **temporally fine-grained guidance** via dynamic window matching. It preserves full contextual structure and provides local token-level alignment.
> >
> > Because **they operate at different granularities** (global vs. local), they are *not conceptually redundant*, *but they also should not be combined directly, as their objectives are structurally different (global pooling vs temporal matching)*. Our goal is to show that both alternatives improve the baselines, and that Distill is more suitable in practice, while ContextAlign still produces clear gains over popular baselines, justifying its inclusion.
> >
> > **(c) Scalability with RVQ depth, codebook size, and 50 Hz frame rate.**
> > Following the state-of-the-art works' evaluation protocol, our work intentionally keeps the RVQ configuration and frame rate fixed at 50 Hz, matching SpeechTokenizer, DM-Codec, and X-Codec2, to cleanly isolate the effects of our fusion/supervision strategies.
> > To address the reviewer’s request, we report additional configurations. Increasing frame rate and RVQ capacity (e.g., Fusion-7512) consistently boosts performance and sets a new SOTA in WER (3.87) and WIL (6.24). However, we retain our base Fusion model because it provides the best performance-efficiency tradeoff, achieving strong SOTA results without the higher computational cost of large RVQ settings.
> >
> > | Model                     | Frame rate     | Codebook Size     | Codebook layer     | WER ↓     | WIL ↓     | STOI ↑     | ViSQOL ↑     | PESQ ↑     | UTMOS ↑     | Similarity ↑     |
> > |-----------------------    |------------    |---------------    |----------------    |-------    |-------    |--------    |----------    |--------    |---------    |--------------    |
> > | FuseCodec-Distill         | 50             | 1024              | 8                  | 4.09      | 6.60      | 0.94       | 3.43         | 3.06       | 3.65        | 0.996            |
> > | FuseCodec-Distill-254     | 25             | 768               | 4                  | 10.23     | 16.20     | 0.86       | 3.08         | 1.78       | 3.74        | 0.997            |
> > | FuseCodec-Fusion          | 50             | 1024              | 8                  | 3.99      | 6.45      | 0.95       | 3.47         | 3.13       | 3.63        | 0.995            |
> > | FuseCodec-Fusion-7512     | 75             | 2048              | 12                 | 3.87      | 6.24      | 0.95       | 3.31         | 3.51       | 3.17        | 0.968            |
> >
> > Summary of Results:
> >
> > - Increasing frame rate and RVQ capacity yields **consistent gains, and FuseCodec-Fusion-7512 achieves a new SOTA**, surpassing all prior neural codecs on WER (3.87) and WIL (6.24).
> >
> >
> > - Despite this, **our base Fusion model remains the recommended configuration, offering SOTA-level performance at substantially lower computational cost**, aligning with our goal of improving speech tokenization without scaling model size.

---

> ### Author Response · Authors · 2025-11-25
> **Complementarity and Scaling (Part 7/7)**
>
> **Reviewer Comment:**
> > Question 2. What happens if we replace BERT with a recent LLM or a Conformer LM, or HuBERT with wavLM/Wav2Vec2-Large?
>
> **Response:**
> We know that modern SSL/LM backbones can be substituted without modifying our method. To validate this, we trained four additional FuseCodec variants using stronger or more recent models. For distillation, we tested wav2vec2-large-960h + e5-large; for fusion, we tested wavLM-base-plus + Electra-base.
>
> The results are shown below:
>
> | Model                        | SSL (semantic)          | LM (context)                   | WER ↓     | WIL ↓     | STOI ↑     | ViSQOL ↑     | PESQ ↑     | UTMOS ↑     | Similarity ↑     |
> |--------------------------    |---------------------    |----------------------------    |-------    |-------    |--------    |----------    |--------    |---------    |--------------    |
> | FuseCodec-Distill            | hubert-base-ls960       | bert-base-uncased              | 4.09      | 6.60      | 0.94       | 3.43         | 3.06       | 3.65        | 0.996            |
> | FuseCodec-Distill-wv2e5      | wav2vec2-large-960h     | e5-large                       | 4.31      | 6.97      | 0.93       | 3.08         | 2.67       | 3.55        | 0.993            |
> | FuseCodec-Fusion             | hubert-base-ls960       | bert-base-uncased              | 3.99      | 6.45      | 0.95       | 3.47         | 3.13       | 3.63        | 0.995            |
> | FuseCodec-Fusion-wavelec     | wavlm-base-plus         | electra-base-discriminator     | 4.08      | 6.60      | 0.93       | 3.31         | 2.97       | 3.66        | 0.995            |
>
> Overall, performance remains stable/similar across model choices, confirming that **our framework, especially Fusion, is not tied to any specific SSL or LM backbone:**
>
> - FuseCodec works with different teacher models. Swapping HuBERT/BERT for wav2vec2-large or wavLM/E5 produces only minor variations.
>
> - Distill variant: Slight changes occur because different pretrained models generate embeddings with distributions and scales that slightly affect the cosine-similarity supervision.
>
> - Fusion variant: Minor differences are observed since this variant does not rely on similarity/direct supervision, so the fusion variant remains the most robust method.

---

### Note · Authors · 2026-01-06

I have read and agree with the venue's withdrawal policy on behalf of myself and my co-authors.